# MT4-MMP deficiency increases patrolling monocyte recruitment to early lesions and accelerates atherosclerosis

Cristina Clemente[1], Cristina Rius[2,3], Laura Alonso-Herranz[4], Mara Martín-Alonso[1], Ángela Pollán[1], Emilio Camafeita [5], Fernando Martínez [6], Rubén A. Mota[1], Vanessa Núñez[4], Cristina Rodríguez[3,7], Motoharu Seiki[8], José Martínez-González[3,9], Vicente Andrés[2,3], Mercedes Ricote[4] & Alicia G. Arroyo [1,10]

Matrix metalloproteinases are involved in vascular remodeling. Little is known about their immune regulatory role in atherosclerosis. Here we show that mice deficient for MT4-MMP have increased adherence of macrophages to inflamed peritonea, and larger lipid deposits and macrophage burden in atherosclerotic plaques. We also demonstrate that MT4-MMP deficiency results in higher numbers of patrolling monocytes crawling and adhered to inflamed endothelia, and the accumulation of Mafb+ apoptosis inhibitor of macrophage (AIM)+ macrophages at incipient atherosclerotic lesions in mice. Functionally, MT4-MMP-null Mafb+AIM+ peritoneal macrophages express higher AIM and scavenger receptor CD36, are more resistant to apoptosis, and bind acLDL avidly, all of which contribute to atherosclerosis. CCR5 inhibition alleviates these effects by hindering the enhanced recruitment of MT4-MMP-null patrolling monocytes to early atherosclerotic lesions, thus blocking Mafb+AIM+ macrophage accumulation and atherosclerosis acceleration. Our results suggest that MT4-MMP targeting may constitute a novel strategy to boost patrolling monocyte activity in early inflammation.

[1] Matrix Metalloproteinases in Angiogenesis and Inflammation Group, Centro Nacional de Investigaciones Cardiovasculares Carlos III (CNIC), Melchor Fernández Almagro 3, 28029 Madrid, Spain. [2] Molecular and Genetic Cardiovascular Pathophysiology Group, Centro Nacional de Investigaciones Cardiovasculares Carlos III (CNIC), Melchor Fernández Almagro 3, 28029 Madrid, Spain. [3] CIBER de Enfermedades Cardiovasculares (CIBERCV), Madrid, Spain. [4] Nuclear Receptor Signaling Group, Centro Nacional de Investigaciones Cardiovasculares Carlos III (CNIC), Melchor Fernández Almagro 3, 28029 Madrid, Spain. [5] Proteomics Unit, Centro Nacional de Investigaciones Cardiovasculares Carlos III (CNIC), Melchor Fernández Almagro 3, 28029 Madrid, Spain. [6] Bioinformatics Unit, Centro Nacional de Investigaciones Cardiovasculares Carlos III (CNIC), Melchor Fernández Almagro 3, 28029 Madrid, Spain. [7] Institut de Recerca del Hospital de la Santa Creu i Sant Pau-Programa ICCC, IIB-Sant Pau, Sant Antoni Maria Claret 167, 08025 Barcelona, Spain. [8] Division of Cancer Cell Research, Institute of Medical Science, University of Tokyo, 4-6-1 Shirokanedai, Minato-ku, Tokyo 108-8639, Japan. [9] Instituto de Investigaciones Biomédicas de Barcelona (IIBB-CSIC), IIB-Sant Pau, Rosselló 161, 08036 Barcelona, Spain. [10] Present address: Centro de Investigaciones Biológicas (CIB-CSIC), Ramiro de Maeztu 9, 28040 Madrid, Spain. Correspondence and requests for materials should be addressed to A.G.A. (email: agarroyo@cnic.es)

Atherosclerosis (AT) is a chronic inflammatory disease with local manifestations in the vasculature in the form of complex multi-cellular atherosclerotic lesions[1]. Cells of the innate immune response, particularly monocytes recruited by the inflamed endothelium and their derived macrophages in the plaque, are essential contributors to the initiation, progression, and eventual rupture of atherosclerotic lesions by mechanisms including the engulfment of LDL particles and the formation of lipid-overloaded foam cells[2,3]. However, the view that inflammation derives from recruited monocytes differentiating into macrophages in the plaque has been challenged and a more complex scenario is envisaged[4]. Macrophages differentiate from monocytes in response to micro-environmental stimuli in the plaque, giving rise to distinct subsets that either contribute to inflammation or favor resolution[5]. Macrophage heterogeneity within atherosclerotic lesions has attracted interest due to its possible therapeutic implications. In mice, macrophages in early atherosclerotic plaques are mainly derived from recruited monocytes, whereas macrophage proliferation is more important in advanced plaques[6]. It remains unclear how macrophage heterogeneity is regulated and contributes to AT initiation and progression. The two main circulating monocyte populations, classical monocytes ($Ly6C^{high}$ in the mouse) and patrolling monocytes ($Ly6C^{low}$), are recruited to the inflamed aorta via distinct chemokine-receptor pathways[7,8]. Classical monocytes have been assigned proinflammatory functions in AT, whereas patrolling monocytes have been considered to have a protective, pro-resolution role; however, it remains unclear how these monocyte populations, particularly patrolling monocytes, contribute to distinct macrophage subsets in the plaque and to AT progression[8].

Macrophages are central actors in the vascular remodeling associated with plaque progression. This vascular remodeling involves key actions of matrix metalloproteinases (MMPs) on the extracellular matrix and vascular smooth muscle cells[9,10]. Less is known, however, about the role of MMPs in regulating inflammatory cells in AT. MT4-MMP (also named MMP17) is anchored to the plasma membrane through a glycophosphatidylinositol anchor that confers distinct features to this protease by positioning it at enriched lipid membrane domains[11,12]. MT4-MMP is expressed by macrophages but its role in inflammation is ill-defined[13,14]. MT4-MMP was first proposed to process pro-TNFα, but peritoneal macrophages lacking MT4-MMP release normal amounts of TNFα[15], indicating alternative functions of MT4-MMP in inflammation. We recently reported that MT4-MMP-null mice are more prone to angiotensin II-induced thoracic aorta aneurysms and to neointima formation in response to carotid ligation[16]. Whether MT4-MMP has roles in inflammatory vascular pathologies such as AT has not been explored.

Here we show that the crawling of patrolling monocytes on the inflamed endothelium is regulated by MT4-MMP-dependent cleavage of αM integrin (Itgam/CD11b), and that this monocyte subset contributes to the accumulation of Mafb+apoptosis inhibitor of macrophage (AIM)+-expressing macrophages in incipient plaques and to overall AT progression. As patrolling monocytes have beneficial effects in infections and the prevention of lung metastasis[8,17], our results also suggest the therapeutic potential of boosting patrolling monocyte activity through MT4-MMP targeting.

## Results

**MT4-MMP-null macrophages accumulate in atherosclerotic plaques**. The peritoneal cavity contains two main macrophage subsets: resident or large peritoneal macrophages (LPM; $CD11b^{hi}/F4/80^{hi}$) and inflammation-induced/monocyte-derived small peritoneal macrophages (SPM; $CD11b^{med}F4/80^{lo/med}$)[18]

(Fig. 1a). MT4-MMP is not expressed in resident LPM[15] but becomes expressed in SPM, the main population present 72 h after thioglycollate (TG) injection (Supplementary Fig. 1a); the function of MT4-MMP in SPMs remains unknown. Although we observed no differences in the number of resident macrophages, fewer SPMs were collected 72 h after TG injection in the peritoneal lavage of MT4-MMP-null mice compared with wild types (Fig. 1b) in spite of similar messenger RNA levels of chemokine receptors[19] (Supplementary Fig. 1b). In parallel, significantly more macrophages were found adhered to the inflamed MT4-MMP-null peritoneum (Fig. 1c). This hyper-adhesive phenotype correlated with elevated cell-surface expression of αM integrin (CD11b/Itgam/Mac-1) in SPMs as assessed by flow cytometry (Fig. 1d). There was no change in *Itgam* mRNA expression (Fig. 1e). Significantly more C3-opsonized sheep erythrocytes were bound to MT4-MMP-deficient SPM than to wild-type cells, confirming αM integrin activity (Fig. 1f). Moreover, MT4-MMP-deficient macrophages covered a significantly larger area than wild-type cells after adhesion for 24 h to the αM integrin ligand fibrinogen, indicating accelerated cell spreading, a post-ligand binding event; this difference was abolished by the specific anti-αM integrin inhibitory antibody M1/70 (Fig. 1g).

The hyper-adhesion of MT4-MMP-null macrophages to the inflamed peritoneum prompted us to explore their behavior in other inflammatory contexts such as the inflamed aortic vessel wall during AT. To test this, low-density lipoprotein receptor-null ($Ldlr^{-/-}$) mice were irradiated and transplanted with bone marrow (BM) cells from either wild-type or MT4-MMP-null mice. We observed proper engraftment and no differences in BM and blood cell populations after 4 weeks on a normal diet (Supplementary Fig. 2a, b). Transplanted mice were then placed on a high-fat diet (HFD) for different periods; weight gain and serum biochemical parameters (glucose, triglycerides, and cholesterol) were similar in both groups throughout the experiment (Supplementary Fig. 2c,d). First, we confirmed that MT4-MMP was expressed in macrophages in the aortic sinus of $Ldlr^{-/-}$ mice transplanted with MT4-MMP-null BM cells (MT4-MMP$^{lacZ/lacZ}$ cells) after 1 and 8 weeks fed a HFD (Supplementary Fig. 2e). We next analyzed the macrophage content in atherosclerotic plaques developed 8 weeks after HFD, finding that macrophages (Mac3+) were significantly more abundant in the aortic sinus of $Ldlr^{-/-}$ mice transplanted with MT4-MMP-null BM cells (Fig. 2a). This increased macrophage burden at 8 weeks correlated with significantly larger lipid lesions in the aortic arch after 12 weeks on the HFD, as revealed by en face Red Oil staining (Fig. 2b). Moreover, neointima area in the aortic sinus was comparable between groups at all time points after HFD (Supplementary Fig. 2f). Analysis of Stary classification[20] revealed that $Ldlr^{-/-}$ mice transplanted with MT4-MMP-deficient BM cells had significantly more advanced lesions after 8 and 12 weeks on the HFD than counterparts transplanted with wild-type BM (Fig. 2c, d and Supplementary Fig. 2g); in particular, MT4-MMP-deficient BM recipients tended to have larger necrotic cores and showed significantly thicker fibrous caps after 12 weeks on HFD (Fig. 2e). Similar results were obtained in double $Ldlr$/MT4-MMP-null mice fed a HFD. These mice nearly recapitulated the phenotype of $Ldlr^{-/-}$ mice transplanted with MT4-MMP-null BM, showing an increased AT burden in the aortic arch and more advanced plaques after 16 weeks of HFD (Supplementary Fig. 3a–d). In this mouse model, MT4-MMP protein and mRNA expression in aorta extracts was low in early stages (8 weeks after HFD) but progressively increased in established AT (16 weeks after HFD); a similar difference was observed between human arteries with arterial intimal thickening (early lesions) and those with established atherosclerotic lesions (Supplementary Fig. 4a–c). Of note, in early mouse

atherosclerotic plaques of MT4-MMP$^{+/+}$/$Ldlr^{-/-}$ mice (8 weeks after HFD), MT4-MMP expression was mainly restricted to macrophages (Supplementary Fig. 4d), indicating that MT4-MMP deficiency appears to principally impact AT progression through its effect on these cells.

Macrophage burden in the atherosclerotic plaque is mostly determined by the balance between early monocyte recruitment and late macrophage proliferation/cell death[21]. After 8 weeks of HFD, plaques from $Ldlr^{-/-}$ mice transplanted with MT4-MMP-null BM showed variable macrophage proliferation but no

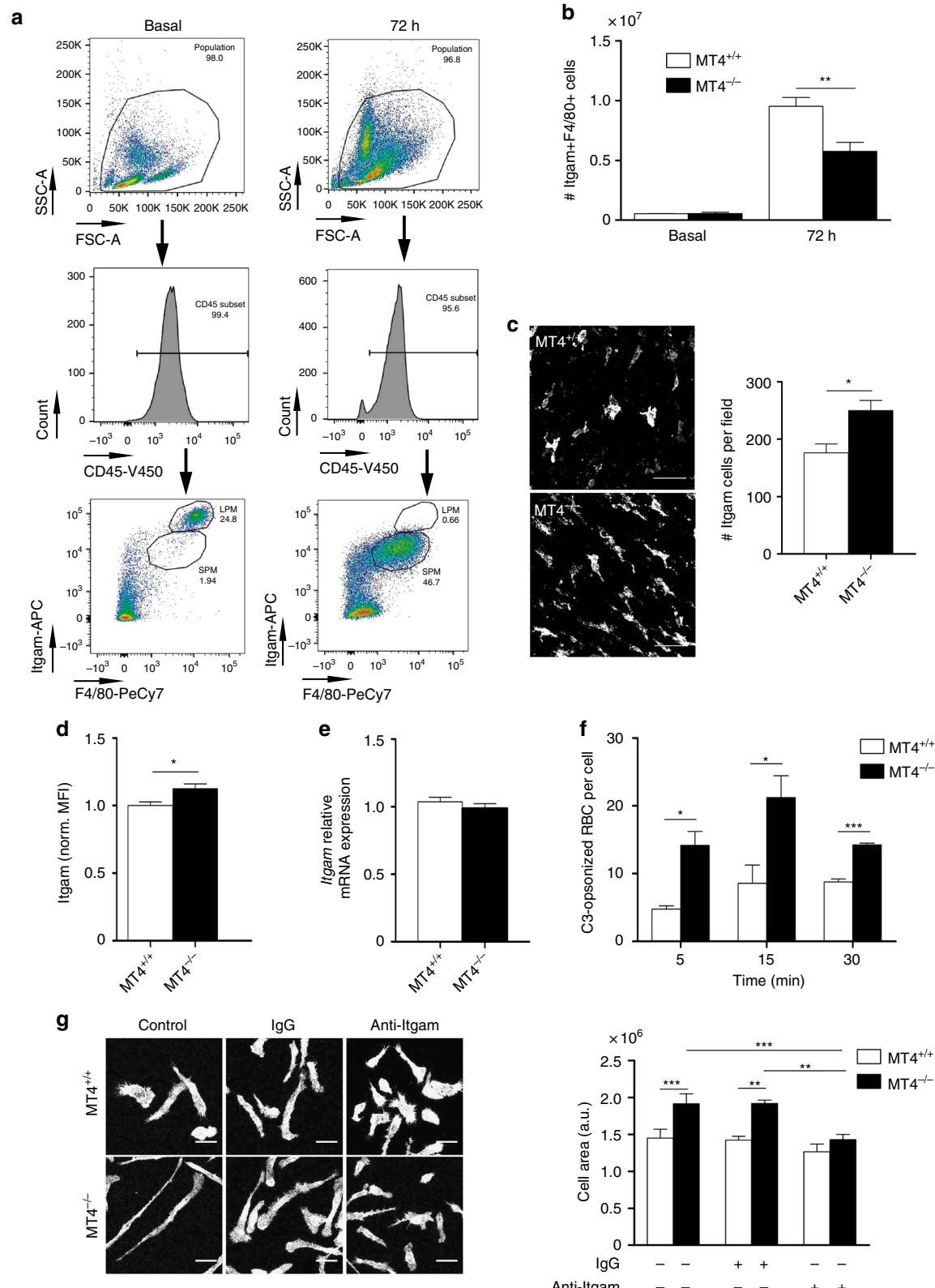

significant increase compared with wild-types ($\sim 33 \pm 21\%$ versus $\sim 25 \pm 10\%$ Mac3+/Ki67+ cells), and cell apoptosis was a rare event (< 1% cleaved caspase-3-positive cells). We also assessed a possible influence of MT4-MMP absence in macrophage egression from inflamed tissues. TG-stimulated macrophages nearly disappeared from the peritoneal cavity in response to lipopolysaccharide (LPS), regardless the genotype with minor contribution of local death and similar numbers found of wild-type and MT4-MMP-null egressed macrophages in the spleen (Supplementary Fig. 5). These data argued in favor of enhanced monocyte recruitment during early AT as the main contributor to increased macrophage burden.

**MT4-MMP absence boosts patrolling monocyte crawling.** Blood monocyte abundance was similar in $Ldlr^{-/-}$ mice transplanted with MT4-MMP-null or wild-type BM cells (1.4% ± 0.5% and 1.3% ± 0.2%, respectively); however, before starting the HFD, lack of MT4-MMP was associated with a significantly lower percentage of circulating patrolling $Ly6C^{low}$ monocytes and these cells displayed increased cell surface levels of αM integrin, similar to MT4-MMP-null SPMs (Figs 3a and 1c).

Increased cell-surface αM integrin in MT4-MMP-null patrolling monocytes and SPMs (Figs 3a and 1d), with no changes in mRNA expression, pointed to posttranslational regulation by MT4-MMP. Both MT4-MMP and αM integrin were located in lipid-rich domains and MT4-MMP-null peritoneal macrophages contained significantly higher levels of αM integrin in these domains (Supplementary Fig. 6a, b). MT4-MMP-null mice were then injected intraperitoneally (i.p.) with lentivirus encoding green fluorescent protein (GFP) and either full-length MT4-MMP or the catalytic dead mutant (E248A) (Fig. 3b). Analysis of peritoneal macrophages at 5 days post injection showed that expression of full-length MT4-MMP yielded normal cell-surface αM integrin levels in infected MT4-MMP-null macrophages (GFP+Itgam+F4/80+); in contrast, infection with the catalytic dead mutant yet resulted in higher surface αM integrin levels than observed in wild-type macrophages, demonstrating that MT4-MMP catalytic activity regulates αM integrin cell-surface levels in vivo (Fig. 3c). We next used the Cleavpredict software application[22] to identify possible direct MT4-MMP cleavage sites in αMβ2 integrin. The predicted and exposed sites were filtered according to the peptide cleavage matrix in the MEROPS database, and candidate cleavage sites were selected in the αM integrin chain (Itgam) between positions 970 and 1000, close to the transmembrane (TM) domain (Fig. 3d and Supplementary Data 1). In silico modeling of interaction between the human αMβ2 integrin heterodimer and the human MT4-MMP dimer confirmed EN$^{977}$LS as the only candidate αM integrin cleavage site accessible to the MT4-MMP catalytic center (Fig. 3d, e). This site is conserved in the mouse αM integrin chain

but not in the related αL integrin chain in humans or mice. Incubation of the human recombinant MT4-MMP catalytic domain with a synthetic peptide R$^{969}$-R$^{987}$ from the human αM integrin sequence resulted in significant cleavage at the αM integrin EN$^{977}$LS site, as assessed by mass spectrometry (Fig. 3f, g).

Crawling of patrolling $Ly6C^{low}$ monocytes on the inflamed endothelium depends on αMβ2 integrin[23] and we therefore next analyzed the in vivo behavior of patrolling monocytes by intravital microscopy of cremaster muscle stimulated with CCL2, an essential chemokine in AT development[24,25]. Only a small number of circulating monocytes, corresponding to patrolling monocytes (CD115+$Ly6C^{low}$), can crawl on the inflamed endothelium[26]. MT4-MMP-null mice had significantly higher numbers of crawling monocytes on CCL2-inflamed endothelium than wild types (Fig. 4a, b and Supplementary Movies 1, 2) with no major differences in crawling velocity, distance traveled, or confinement index (Supplementary Fig. 7a). Similar total number of monocytes (CD115+Ly6G−) were observed rolling or adhered to the CCL2-inflamed endothelium (Fig. 4c), indicating a selective effect of MT4-MMP absence in patrolling monocyte behavior. Moreover, the increased crawling of MT4-MMP-null patrolling monocytes was abolished when αM integrin was blocked with the inhibitory antibody M1/70 (Fig. 4a, b and Supplementary Movies 1–4). MT4-MMP-null mice also had fewer neutrophils rolling on the inflamed endothelium, suggesting their retention by crawling patrolling monocytes, as reported in other contexts (Supplementary Fig. 7b)[23].

**MT4-MMP absence increases Mafb+AIM+ macrophages in early AT.** Low efficiency is a common limitation for the analysis of patrolling monocyte recruitment to the inflamed endothelium of atherosclerotic plaques[21,27]. Still, whole-mount staining showed significantly more patrolling monocytes (CD115+Ly6C−) adhered to the endothelium of the lesser curvature of the aortic arch in MT4-MMP-null BM-transplanted $Ldlr^{-/-}$ mice than in similar mice transplanted with wild-type cells after 3 days on the HFD (Fig. 5a); no between-genotype differences were detected in cell surface levels of CCR5, the chemokine receptor involved in patrolling monocyte recruitment[27] and CCR2 (MFI$_{CCR5}$ = 452 ± 53 and 498 ± 47, and MFI$_{CCR2}$ = 592 ± 65 and 613 ± 75 in patrolling monocytes from wild-type and MT4$^{-/-}$-BMT $Ldlr^{-/-}$ mice, respectively). Classical monocytes (CD115+Ly6C+) were recruited at higher proportion at this incipient stage[21,27] but similarly regardless the genotype (Fig. 5a). Notably, MT4-MMP-null sorted patrolling monocytes adoptively transferred into $Ldlr^{-/-}$ mice also adhered at higher numbers to the lesser curvature of the aorta after 3 days on the HFD compared with wild types confirming the cell autonomous impact of MT4-MMP absence on patrolling monocyte behavior (Fig. 5b and Supplementary Fig. 8).

**Fig. 1** Enhanced trapping of MT4-MMP-null peritoneal macrophages due to increased αM integrin (Itgam) levels and activity. **a** Representative flow cytometry dot plots and histograms of mouse peritoneal macrophages stained for CD45, Itgam, and F4/80 in basal conditions (left) and 72 h after thioglycollate (TG) injection (right). **b** Number of macrophages (Itgam+F4/80+) collected in the peritoneal eluate of wild-type and MT4-MMP-null mice at the indicated times after TG injection; $n = 3$ mice in basal and $n = 12$ mice at 72 h per genotype in one in basal and four in 72 h independent experiments, respectively. **c** Representative confocal microscopy images (left) and quantification (right) of monocytes/macrophages (Itgam+) in the peritoneal membrane 72 h after TG injection; $n = 6$ mice per genotype in two independent experiments; scale bar, 50 μm. **d** Flow cytometry analysis of Itgam cell-surface levels in TG-elicited macrophages (Itgam+F4/80+) obtained 72 h after TG injection; $n = 20$ mice per genotype in four independent experiments. **e** qPCR analysis of Itgam mRNA in TG-elicited macrophages adhered to plastic overnight; $n = 6$ samples per genotype in two independent experiments. **f** Itgam integrin affinity assessed as the number of C3-opsonized red blood cells (RBCs) bound to TG-elicited peritoneal macrophages adhered to glass; $n = 6$ samples per genotype from two independent experiments. **g** Representative fluorescence images of TG-elicited macrophages adhered to fibrinogen for 24 h and labeled for F-actin in the presence or absence of Itgam blocking antibody M1/70 or IgG isotype control (left). The histogram shows quantification of the cell area (right). Scale bar, 30 μm. $n = 6$ samples per genotype in two independent experiments. Data were tested by two-way ANOVA followed by Bonferroni's post test in **f**, by two-tailed Student's $t$-test in **b**, **c**, **d**, and **e**, and by one-way ANOVA followed by Bonferroni's post test in **g**. Results are expressed as mean ± SEM. *$p < 0.05$, **$p < 0.01$, and ***$p < 0.001$

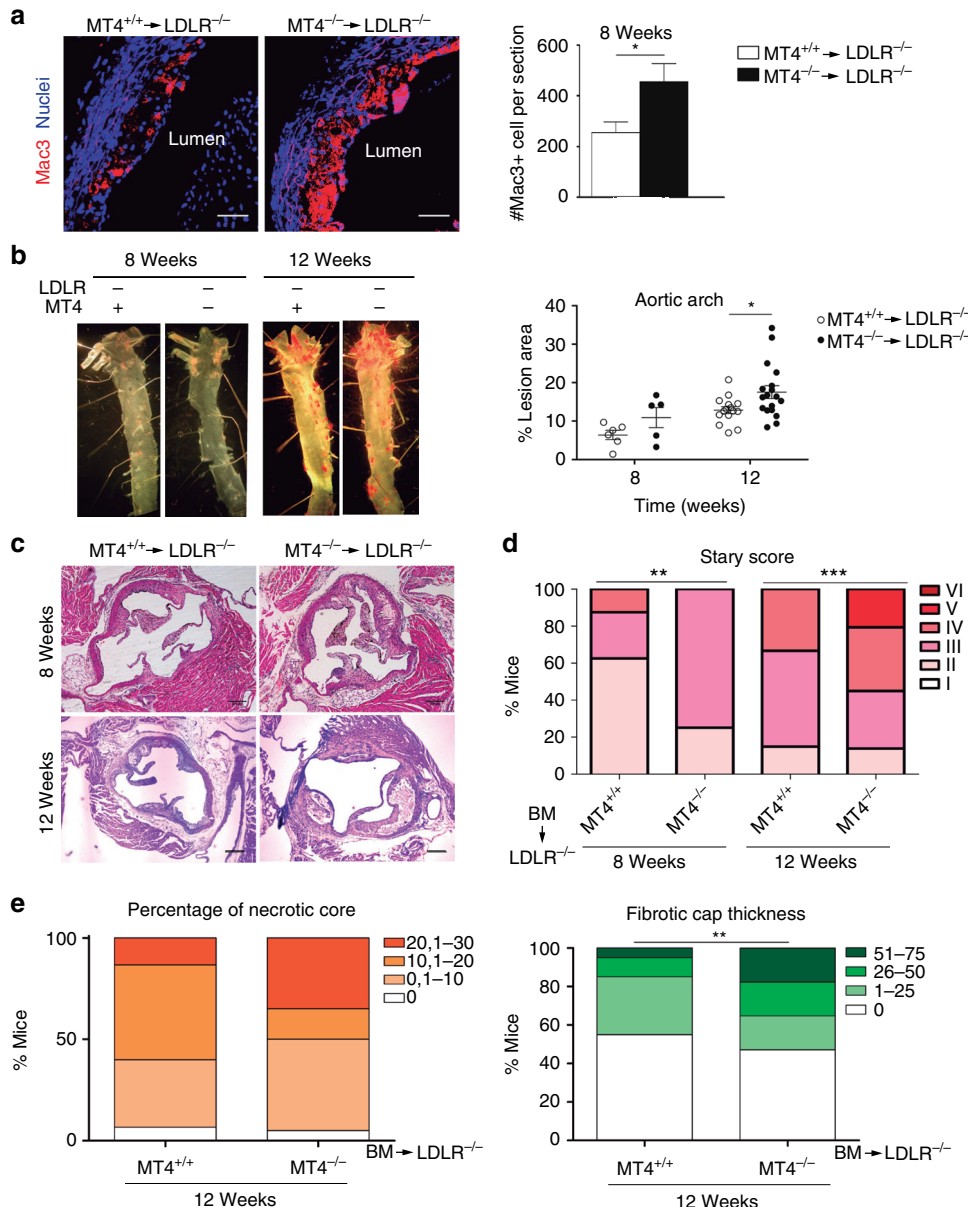

**Fig. 2** Lack of MT4-MMP in BM-derived cells results in increased macrophage burden in atherosclerotic plaques and accelerated AT. **a** Representative images of Mac3 immunostaining (red; nuclei in blue) in transverse sections of aortas from $Ldlr^{-/-}$ mice transplanted with MT4-MMP$^{+/+}$ (MT4$^{+/+}$) or MT4-MMP$^{-/-}$ (MT4$^{-/-}$) BM cells and fed a HFD for 8 weeks; scale bar, 20 μm. The right panel shows Mac3-positive cells quantified by Image J. $n = 6$ mice per genotype in two independent experiments. **b** Representative images of en face Oil Red-stained aortas from BM-transplanted $Ldlr^{-/-}$ and fed a HFD for 8 or 12 weeks (left) and graph shows the area (%) of Oil Red-positive lesions in the aortic arch (right); $n = 6$ and $n = 16$ mice per genotype for 8 and 12 weeks in two and three independent experiments, respectively. **c** Representative images of transverse sections of aortic sinus stained with H&E of BM-transplanted $Ldlr^{-/-}$ mice; scale bar, 200 μm. **d** Stary scoring (I–VI) of aortic lesions of BM-transplanted $Ldlr^{-/-}$ mice, shown as a percentage of all mice for each condition after feeding a HFD for 8 or 12 weeks; $n = 6$ and $n = 16$ mice per genotype and time point in two and three independent experiments. **e** Bar graphs show the percentage of BM-transplanted $Ldlr^{-/-}$ mice for each range of % of necrotic area (left) and fibrotic cap thickness (right) after 12 weeks of HFD; $n = 16$ mice per genotype in three independent experiments. Data were tested by two-tailed Student's $t$-test in **a**, by two-way ANOVA followed by Bonferroni's post test in **b**, and by $\chi^2$-test for a trend in **d**, **e**. Results are expressed as mean ± SEM. *$p < 0.05$, **$p < 0.01$, and ***$p < 0.001$

Although patrolling monocytes express genes related to cholesterol sensing and responses, their contribution to plaque lipid accumulation remains poorly characterized[21]. Analysis of incipient plaques after 7 days on the HFD showed no significant differences in macrophage number between $Ldlr^{-/-}$ mice transplanted with MT4-MMP-null or wild-type cells (Fig. 6a, b). We next explored whether increased recruitment of patrolling monocytes influences macrophage subset composition rather than their abundance.

Patrolling monocytes initiate a macrophage differentiation program in response to infection by upregulating Mafb in the peritoneal cavity[26], and this transcription factor can promote AT[26,28]. After 7 days on the HFD, incipient plaques from MT4$^{-/-}$-transplanted $Ldlr^{-/-}$ mice contained significantly more (~ 3-fold) Mac3+ macrophages with nuclear Mafb (49.6 ± 15%) than those from mice transplanted with wild-type cells (22.7 ± 10%; Fig. 6a, b), and this subset seemed to be more proliferative (~ 1.5-fold) in MT4$^{-/-}$-transplanted $Ldlr^{-/-}$ mice. All

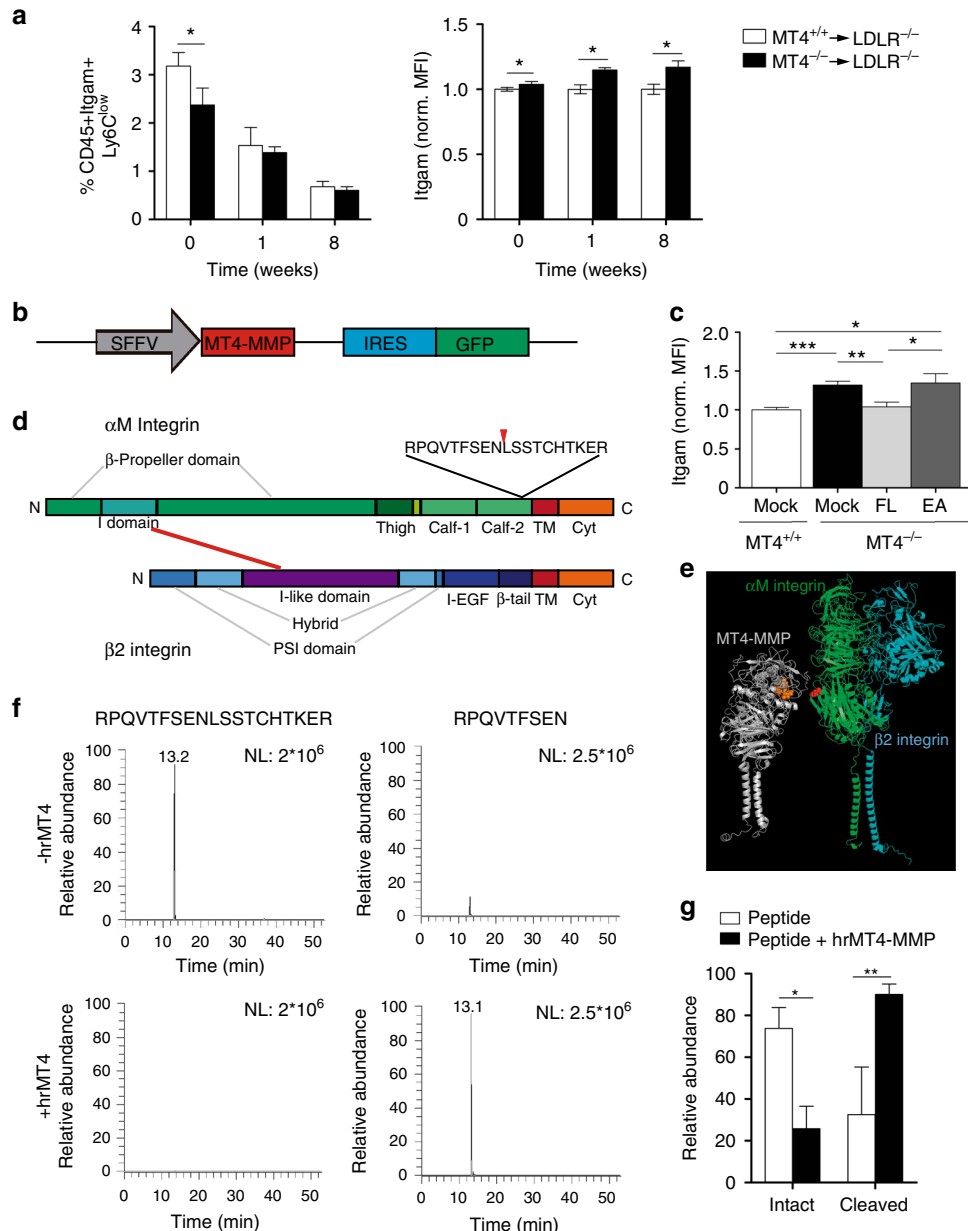

**Fig. 3** The protease MT4-MMP can cleave the αM integrin chain (Itgam). **a** Percentage of circulating patrolling monocytes (CD45+Itgam+Ly6Clow, excluding granulocytes) and the normalized mean fluorescence intensity (MFI) of Itgam cell-surface levels in patrolling monocytes from $Ldlr^{-/-}$ mice transplanted with MT4-MMP$^{+/+}$ or MT4-MMP$^{-/-}$ BM cells and fed a HFD for 0, 1, or 8 weeks; $n = 12$, $n = 6$, and $n = 6$ mice in basal, 1 week and 8 weeks per genotype; four independent experiments in basal and two independent experiments at 1 and 8 weeks. **b** Design of lentiviral (LV) vector with SFFV-driven Mmp17 (MT4-MMP) expression and IRES-driven expression of green fluorescent protein (GFP). **c** LV encoding full-length mouse MT4-MMP (FL), the catalytic inactive mutant (E248A, EA), or GFP only (mock) were i.p. injected into MT4-MMP-null mice. Itgam cell surface levels were assessed by flow cytometry in the infected peritoneal macrophages (GFP+Itgam+F4/80+) 5 days after infection; $n = 6$ mice per condition in two independent experiments. **d** Depiction of human αMβ2 integrin domains, indicating the predicted cleavage site at position 977 in the Calf-2 domain of human αM integrin. **e** In silico model of human MT4-MMP dimer (gray) and αMβ2 integrin (αM chain, green; β2 chain, blue) showing the putative cleavage site between N977 and L978 (red) in the αM chain, and the catalytic active center in the MT4-MMP dimer (orange). **f** Representative extracted ion chromatogram of peptides obtained after incubation of the synthetic human αM integrin peptide RPQVTFSENLSSTCHTKER in the presence or absence of human recombinant MT4-MMP catalytic domain (hrMT4). **g** Quantification of the relative abundance of the intact RPQVTFSENLSSTCHTKER and N-terminal peptide fragments in each condition; $n = 4$ independent experiments. Data were tested by two-way ANOVA followed by Bonferroni's post test in **a**, by one-way ANOVA followed by Bonferroni's post test in **c**, and by two-tailed Student's $t$-test in **g**. IRES, internal ribosome entry site; SFFV, spleen focus-forming virus. Results are expressed as mean ± SEM. *$p < 0.05$, **$p < 0.01$, and ***$p < 0.001$.

macrophages with nuclear Mafb also expressed AIM (apoptosis inhibitor of macrophages; Fig. 6a, b), which promotes survival of lipid-loaded macrophages[29]. In this line, a trend to reduced macrophage apoptosis (Mac3+/cleaved caspase-3+ cells) was

apparent in incipient plaques of MT4$^{-/-}$-transplanted $Ldlr^{-/-}$ mice (~ 6 ± 2% vs ~ 8 ± 3% in those transplanted with wild-type cells). The increase in Mac3+ macrophages expressing AIM in the incipient plaques from MT4$^{-/-}$-transplanted $Ldlr^{-/-}$ mice

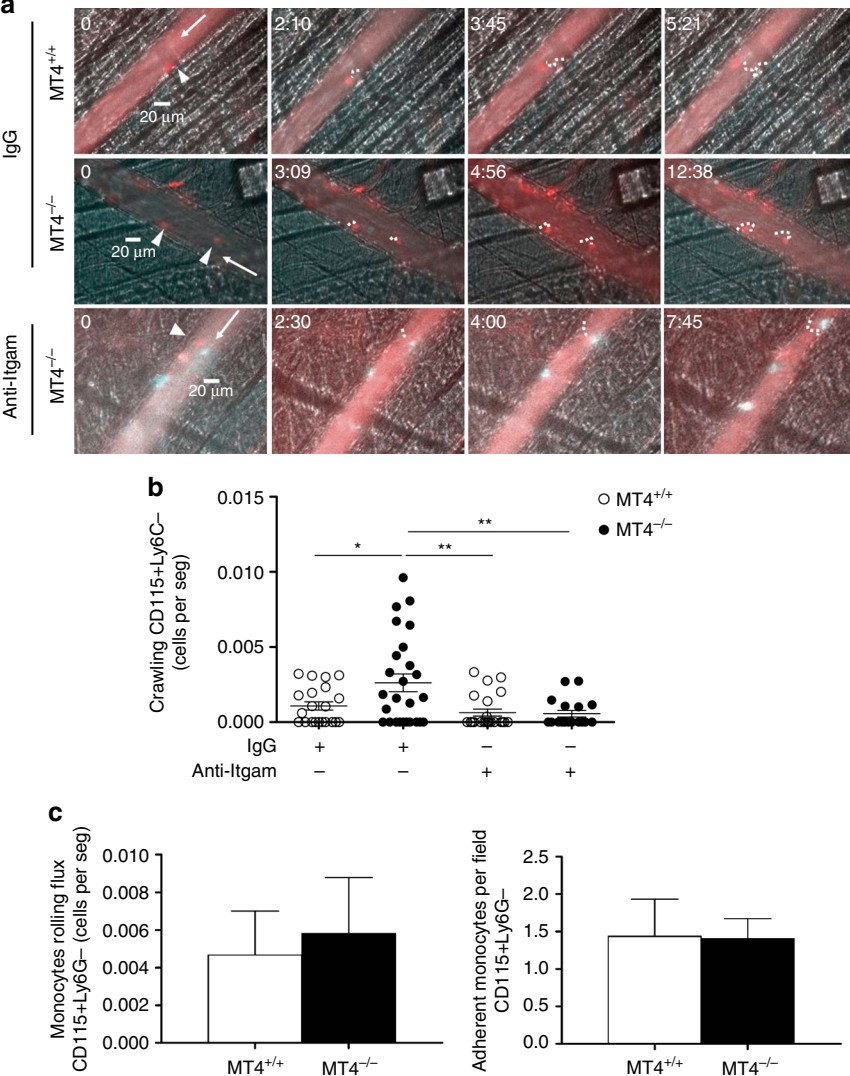

**Fig. 4** Enhanced αM integrin-dependent crawling of MT4-MMP-null patrolling monocytes on CCL2-inflamed endothelium. **a** Representative intravital microscopy images of CD115+/Ly6C– patrolling monocytes (CD115 in red and Ly6C in green) crawling on the CCL2-inflamed endothelium in the cremaster muscle of wild-type (MT4$^{+/+}$) and MT4-MMP-null (MT4$^{-/-}$) mice; the recording was performed in the presence of anti-Itgam blocking antibody (M1/70) or IgG isotype control. Arrowheads, arrows, and dots respectively indicate individual patrolling monocytes, blood flow, and monocyte trajectory. Time of recording is indicated. **b** The graph shows the numbers of crawling patrolling monocytes recorded in **a** in every venule from five independent mice per genotype and condition in two independent experiments. **c** Quantification of CD115+Ly6G- rolling (left) and adherent (right) monocytes in the CCL2-inflamed endothelium in the cremaster muscle of wild-type (MT4$^{+/+}$) and MT4-MMP-null (MT4$^{-/-}$) mice. $n = 8$ mice per genotype in two independent experiments. Data were tested by one-way ANOVA followed by Bonferroni's post test in **b** and by two-tailed Student's $t$-test in **c**. Results are expressed as mean ± SEM.*$p < 0.05$, **$p < 0.01$, and ***$p < 0.001$

(39 ± 6% vs 27.6 ± 9% in those transplanted with wild-type cells) remained up to 8 weeks but was no longer observed in advanced plaques after 12 weeks on a HFD (Supplementary Fig. 9a, b). AIM also participates in oxLDL uptake[30]. Accordingly, ~ 50% of Mafb+ macrophages were positive for adipophilin in wild-type BM-transplanted $Ldlr^{-/-}$ mice and this proportion was significantly higher (~ 70%) in $Ldlr^{-/-}$ mice transplanted with MT4-MMP-null BM cells (Fig. 6c, d).

To further understand the influence of MT4-MMP absence on the functional phenotype of Mafb+AIM+ macrophages, we analyzed TG-elicited peritoneal macrophages[27]. Mirroring the accumulated subset observed in incipient atherosclerotic plaques, significantly more MT4-MMP-null macrophages contained Mafb at the nucleus (~ 60%, threefold) compared with wild types (~ 20%), and they also expressed higher levels of total and cell-surface AIM, detected by immunostaining and flow cytometry (Fig. 7a–c); $Mafb$ and $Cd5l$ ($Aim$) mRNA levels showed no

differences (Supplementary Fig. 10a). Increased AIM expression resulted in significantly lower numbers of apoptotic MT4-MMP-null peritoneal macrophages after cycloheximide treatment (Fig. 7d). AIM also increases CD36-mediated oxLDL uptake and foam-cell formation[30]. Flow cytometry analysis revealed significantly increased cell-surface CD36 expression in TG-elicited MT4-MMP-null macrophages and these cells were more efficient at acLDL binding than wild types (Fig. 7e, f); no differences were detected in $Cd36$ mRNA levels (Supplementary Fig. 10a). Further analysis showed that MT4-MMP-null peritoneal macrophages did not differ from wild types in the mRNA levels of proinflammatory cytokines ($Il1b$, $Tnfa$, and $Il6$) or anti-inflammatory cytokines ($Il10$ and $Tgfb$) (Supplementary Fig. 10b).

**CCR5 inhibition hinders accelerated AT in MT4-MMP absence.** To prove that enhanced recruitment of patrolling monocytes in MT4-null BMT $Ldlr^{-/-}$ mice fed the HFD for 3 days

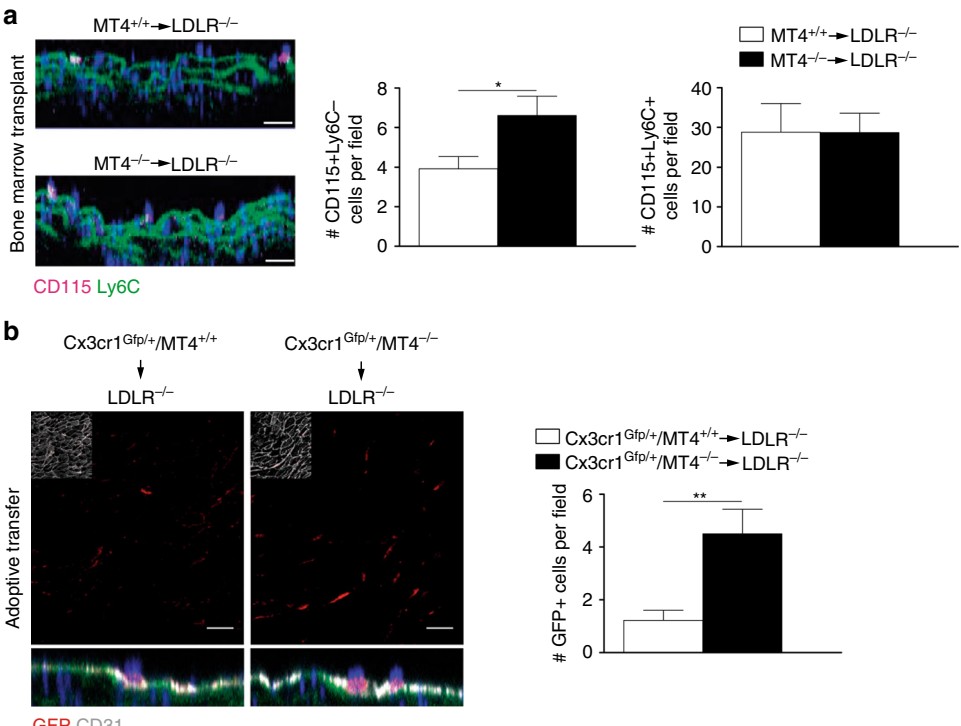

**Fig. 5** Increased recruitment of MT4-MMP-null patrolling monocytes in incipient atherosclerotic lesions. **a** Representative orthogonal XZ view images of whole-mount-stained lesser curvature of the aortic arch from *Ldlr*$^{-/-}$ mice transplanted with MT4-MMP$^{+/+}$ or MT4-MMP$^{-/-}$ BM cells and fed a HFD for 3 days. Samples were stained for CD115 (magenta) and Ly6C (green); elastin autofluorescence (green) and nuclei (Hoechst, blue). The bar graph (right) shows the quantification of the number of patrolling (CD115+Ly6C−) and classical monocytes (CD115+Ly6C+) in the aorta lumen; $n = 6$ mice per genotype in two independent experiments. **b** Representative confocal microscopy images of whole-mount-stained lesser curvature of the aortic arch from *Ldlr*$^{-/-}$ mice adoptively transfer with Cx3cr1$^{Gfp/+}$ MT4-MMP$^{+/+}$ or Cx3cr1$^{Gfp/+}$ MT4-MMP$^{-/-}$ patrolling monocytes and fed a HFD for 3 days. Samples were stained for GFP (red) and CD31 (gray); elastin autofluorescence (green) and nuclei (Hoechst, blue). A z-stack of the confocal microscopy sections close to the lumen (with an inset of CD31 staining) is shown to the top and the orthogonal XZ view of the merged images to the bottom. Scale bar, 20 µm. The bar graph (right) shows the quantification of the number of transferred monocytes (GFP+) in the aorta lumen; $n = 9$ mice per genotype in two independent experiments. Data were tested by Student's *t*-test. Results are expressed as mean ± SEM.*$p < 0.05$, **$p < 0.01$

(Fig. 5a) was promoting Mafb+AIM+ macrophage accumulation and AT acceleration at later stages (Figs. 6a, b and 2), we sought to selectively block patrolling monocyte recruitment. For that, we used Maraviroc (MRV), a CCR5 antagonist used in the clinic[31], as patrolling monocyte preferentially employ this receptor to enter atherosclerotic plaques[27]. Daily treatment of mice with MRV reduced the low numbers of patrolling monocytes[27] consistent with decreased recruitment and abolished the enhanced adherence of patrolling monocytes to the aorta of MT4-null BM-transplanted *Ldlr*$^{-/-}$ mice after 3 days on the HFD (Fig. 8a). Concomitantly, MRV also led to loss of the increased abundance in Mafb+AIM+ macrophages at 7 days and in total macrophages at 8 weeks on the HFD in MT4-null BMT *Ldlr*$^{-/-}$ mice overall delaying AT progression, as shown by quantification of lipid deposits, neointima area, and Stary score, to similar levels than those observed in MRV-treated *Ldlr*$^{-/-}$ mice transplanted with wild-type cells[32] (Fig. 8b–f and Supplementary Fig. 11a–c).

The data obtained with the CCR5 antagonist demonstrate that the atherosclerotic phenotype observed in the absence of MT4-MMP is related to its primary impact on the enhanced early recruitment of patrolling monocytes to the inflamed aorta.

## Discussion
Patrolling monocytes exert their surveillance activity within the vasculature, where they recognize endothelial damage and promote repair[23]. A vascular protective function for patrolling monocytes in inflammation was supported by the exacerbated AT

in mice lacking the orphan nuclear receptor Nur77/Nr4a1, which present a dramatic reduction in Ly6C$^{low}$ patrolling monocyte production[2,8,33]. These mice have, however, other defects such as a proinflammatory shift of Nur77-null macrophages, which could in part mediate the observed phenotype[33]. Moreover, a separate study did not detect changes in lipid lesions in *Ldlr*-null mice transplanted with Nur77$^{-/-}$ BM[34], and therefore debate has persisted about the atheroprotective role of patrolling monocytes. Our findings in MT4-MMP-null mice provide the first evidence for a contribution of patrolling monocytes to a proatherogenic Mafb+AIM+ macrophage subset in early plaques. The contribution of patrolling monocytes to AT may also be clarified by a recent mouse model targeting a super-enhancer that ablates Nur77-dependent Ly6C$^{low}$ monocytes, while preserving Nur77 expression in tissue macrophages and macrophage responses to inflammation[35].

Macrophage burden mainly depends on monocyte recruitment in early plaques and on local macrophage proliferation in advanced plaques[6,36]. Established atherosclerotic lesions in MT4-MMP-null transplanted HFD-fed *Ldlr*$^{-/-}$ mice had an increased macrophage abundance unrelated to differences in proliferation. Although Ly6C$^{low}$ patrolling monocytes are recruited early to incipient plaques in a CCR5-dependent manner[37], the mechanisms that govern their crawling, adherence, and recruitment to the plaque remain poorly defined[8]. Our study identifies MT4-MMP-mediated cleavage of αM integrin as a mechanism for fine-tuned regulation of patrolling monocyte crawling. MT4-MMP

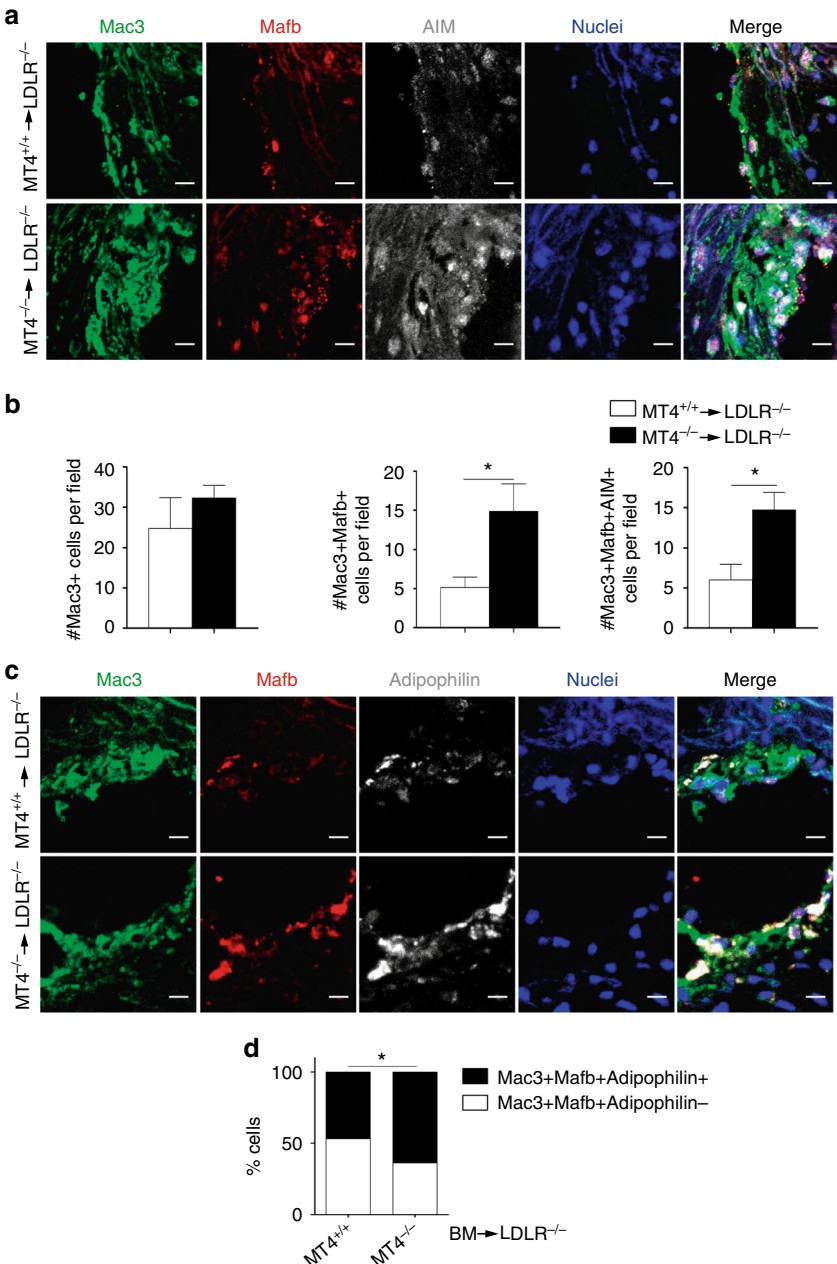

**Fig. 6** Lack of MT4-MMP in patrolling monocytes leads to the accumulation of Mafb+AIM+ macrophages in incipient atherosclerotic plaques. **a** Representative images of transverse sections of aortic sinus from $Ldlr^{-/-}$ mice transplanted with MT4-MMP$^{+/+}$ (MT4$^{+/+}$) or MT4-MMP$^{-/-}$ (MT4$^{-/-}$) BM cells and fed a HFD for 1 week; sections were labeled for Mac3 (green), Mafb (red), and AIM (white), and with Hoechst (blue; nuclei); scale bar, 10 μm. **b** Number of Mac3+ cells (left), Mac3+Mafb+ cells (middle), and Mac3+Mafb+AIM+ cells (right) in the plaques of BM-transplanted $Ldlr^{-/-}$ mice fed a HFD for 1 week. **c** Representative images of transverse sections of aortic sinus from BM-transplanted $Ldlr^{-/-}$ mice fed a HFD for 1 week; sections were labeled for Mac3 (green), Mafb (red), and adipophilin (white), and with Hoechst (blue; nuclei); scale bar, 10 μm. **d** Relative % of adipophilin-positive and adipophilin-negative cells within the Mac3+Mafb+ population of BM-transplanted $Ldlr^{-/-}$ mice (1 week on HFD); $n = 7$ mice per genotype in two independent experiments. Data were tested by two-tailed Student's $t$-test in **b** and by Fisher's exact test in **d**. Results are expressed as mean ± SEM. *$p < 0.05$

absence thus resulted in enhanced adhesion of patrolling monocytes on the inflamed endothelium and therefore their recruitment to incipient atherosclerotic plaques in a cell autonomous manner as shown by adoptive transfer experiments. Attention has also focused on macrophage heterogeneity in the atherosclerotic plaque, with new subsets being characterized, adding Mox, Mhem, and M4 macrophages to the established M1 and M2 categories[5]. As patrolling monocytes express genes for cholesterol sensing and response[21], it is of particular interest to determine whether their recruitment to the plaque influences macrophage

composition, lipid handling, and overall AT progression. In this regard, patrolling monocytes are more prone to develop into CD11c+ cells in the plaque, but the function of these CD11c+ cells in AT was not elucidated[27,38]. We confirmed the presence of more CD11c+ cells in incipient plaques from MT4$^{-/-}$-transplanted $Ldlr^{-/-}$ mice (~ 2-fold), but these cells contained fewer lipids than CD11c+ cells in mice transplanted with wild-type BM, suggesting a limited contribution to the observed atherosclerotic phenotype (data not shown). Macrophages in MT4-MMP-null-derived plaques had an elevated accumulation of

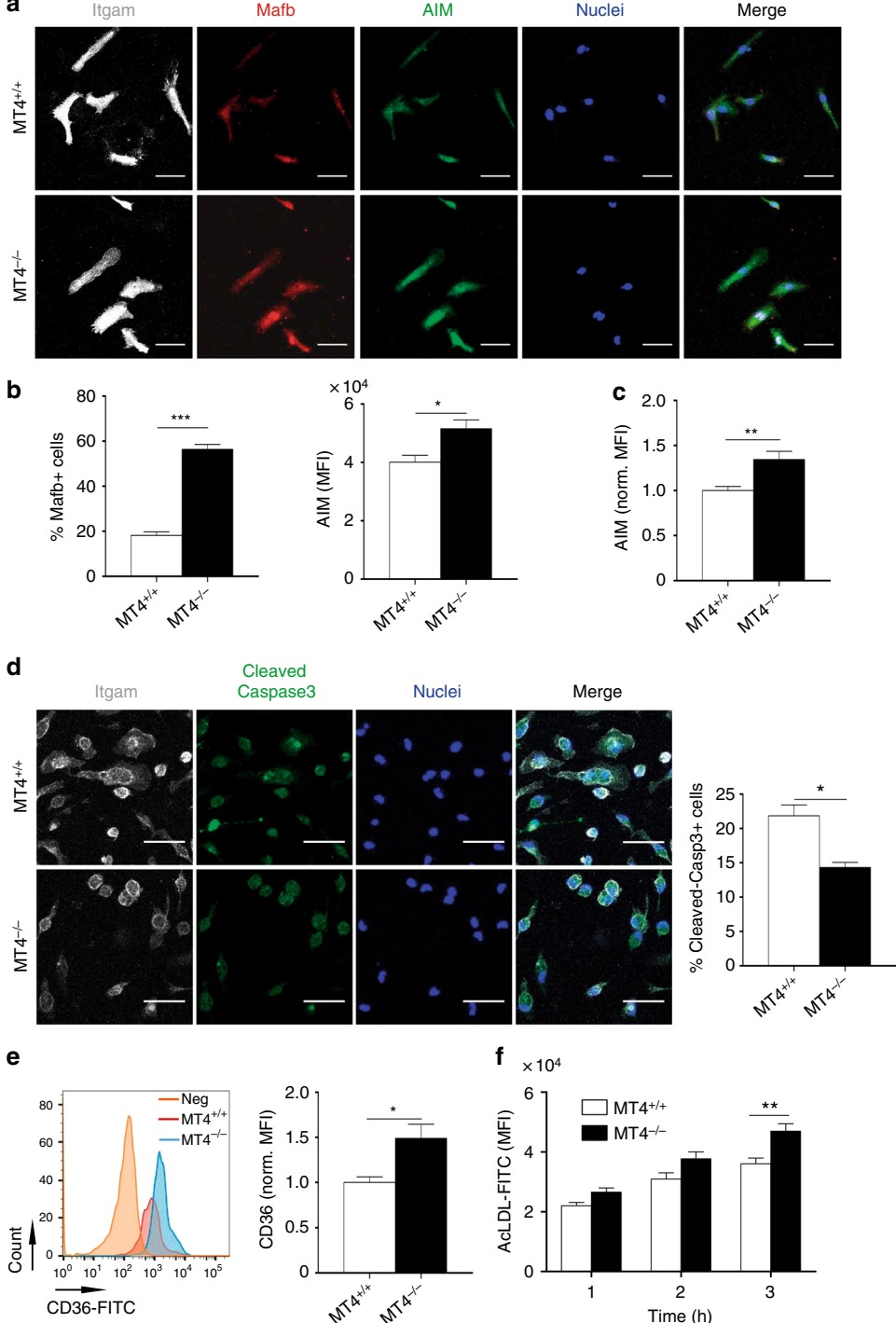

**Fig. 7** MT4-MMP-null Mafb+/AIM+ peritoneal macrophages exhibit above-normal CD36 at the cell surface and enhanced acLDL binding. **a** Representative images of MT4-MMP$^{+/+}$ (MT4$^{+/+}$) or MT4-MMP$^{-/-}$ (MT4$^{-/-}$) mouse peritoneal macrophages elicited by 72 h TG stimulation and labeled for Itgam (white), Mafb (red), and AIM (green), and with Hoechst (blue, nuclei); scale bar, 20 μm. **b** Graphs show the percentage of cells Mafb+ in the nuclei (left) and the MFI of AIM within Mafb+ cells (right); $n = 6$ in two independent experiments. **c** Quantification of normalized MFI of AIM analyzed by flow cytometry in MT4$^{+/+}$ or MT4$^{-/-}$ mouse peritoneal macrophages elicited by 72-hour TG stimulation; $n = 6$ in two independent experiments. **d** Representative images of TG-elicited MT4$^{+/+}$ or MT4$^{-/-}$ mouse peritoneal macrophages treated with cycloheximide (100 μg ml$^{-1}$) for 6 h and labeled for Itgam (white) and cleaved-caspase 3 (green) and Hoechst (blue, nuclei); scale bar, 20 μm. Quantification of the percentage of cleaved-caspase 3-positive cells is shown on the right; $n = 3$ in one experiment. **e** Representative flow cytometry histogram plot of CD36 staining in TG-elicited MT4$^{+/+}$ or MT4$^{-/-}$ mouse peritoneal macrophages (left) and quantification of normalized MFI (right); $n = 6$ in two independent experiments. **f** MFI of AcLDL-FITC binding to TG-elicited MT4$^{+/+}$ or MT4$^{-/-}$ mouse peritoneal macrophages for the indicated times; $n = 6$ in two independent experiment. Data were tested by Student's $t$-test in **b**, **c**, **d**, and **e**, and by two-way ANOVA followed by Bonferroni's post test in **f**. Results are expressed as mean ± SEM. *$p < 0.05$, **$p < 0.01$, and ***$p < 0.001$

nuclear Mafb similar to observations in other contexts, thus supporting the idea that Mafb may be an intrinsic signature of patrolling monocyte differentiation into macrophages, regardless the inflammatory context[26]. Mafb+ lipid-loaded macrophages have been reported in mouse and human established athero-sclerotic plaques, but the origin of this subset was not eluci-dated[28]. Mac3+Mafb+ macrophages also expressed AIM (apoptosis inhibitor of macrophages), which by inhibiting foam-cell apoptosis and favoring oxLDL uptake[28–30] may underlie the larger lipid lesions developed in the aortas of MT4[−/−]-trans-planted *Ldlr*[−/−] mice after 12 weeks on the HFD. Data obtained with the CCR5 inhibitor confirmed the dependence of patrolling monocytes on this receptor for their efficient recruitment to the plaque and the existence of non-CCR5 factors whose functional significance remains to be determined[27]. CCR5 inhibitor data also argue in favor of increased early recruitment of patrolling monocytes to athero-prone aorta areas as the primary contributor to Mafb+AIM+ macrophage abundance and AT acceleration in the absence of MT4-MMP. Notably, the phenotype was not restricted to the atherosclerotic context, since we also observed Mafb+AIM+ macrophage accumulation in TG-elicited MT4-MMP-null peritoneal macrophages. Furthermore, these macro-phages exhibited higher levels of cell-surface AIM and of CD36 (a gene expressed by patrolling monocytes[21]) and bound acLDL more avidly[39], in line with the role assigned to AIM in favoring CD36-mediated oxLDL uptake and foam-cell formation[30].

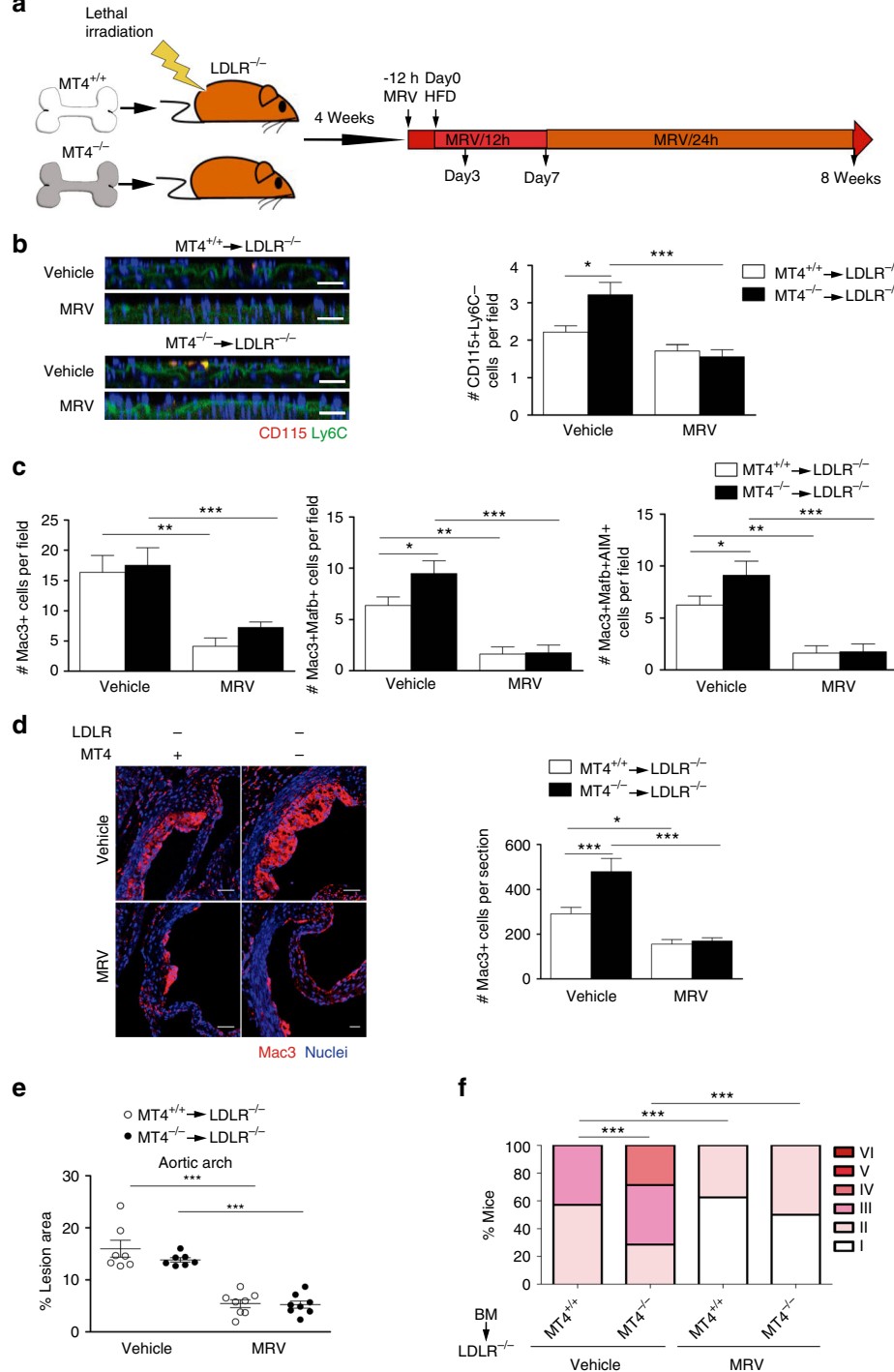

Interestingly, the functional outcome of high CD36 levels in MT4-MMP-null Mac3+Mafb+AIM+ macrophages may be further shaped by environmental factors: in a lipid-rich milieu like AT, CD36 expression would favor disease progression; however, in contexts such as bacterial infection or amyloid deposition, CD36 might promote phagocytosis and resolution[40]. The Mac3+Mafb+AIM+ subset seems to be unique within the multi-dimensional model of macrophage plasticity[41], with cytokine profiling indicating no shift to a proinflammatory or antiinflammatory phenotype; however, amplified αM integrin-mediated signaling in MT4-MMP-null macrophages may still produce changes in certain cytokines, such as CXCL2[42]. Cooperation of αM integrin with CD36 and TLR2 in cell-surface lipid domains on macrophages has been shown to promote lipid accumulation and downmodulate the inflammatory response[43]; future studies will elucidate whether this cooperation takes place in Mac3+Mafb+AIM+ macrophages, particularly in the absence of MT4-MMP. Our results identify a unique apoptosis-resistant, lipid-prone, and non-inflammatory Mac3+Mafb+AIM+ subset that contributes to AT progression; the relation of this subset to subsets previously reported in the atherosclerotic plaque (Mox, M (Hem), M(Hb), and M4)[5] deserves further analysis. In light of these results, combined strategies targeting Mafb+ and Mafb− macrophages may prove to be more effective at ameliorating AT progression. Determining whether patrolling monocyte-derived macrophages still have a pro-resolution role at later stages in established AT lesions would require the use of inducible depletion strategies.

This report also describes the first MT4-MMP function in monocytes/macrophages unrelated to the previously proposed pro-TNFα processing[15,13,14]. We identify the αM integrin chain (Itgam), present in lipid membrane microdomains, as a novel MT4-MMP substrate[12,44,45], and the lentiviral strategy indicates that MT4-MMP catalytic activity regulates αM integrin cell surface levels in patrolling monocytes also in vivo. Serine proteases and MMP-9 cleave the β2 integrin chain[46,47] and this shedding is thought to allow leukocyte detachment after endothelial adhesion[48,49]. We identified an MT4-MMP cleavage site at position $N^{977}$ of the αM integrin chain, which is conserved in human and mouse αM integrin but not in the related αL integrin chains of either species. In response to vessel injury or inflammation, patrolling monocytes change their crawling pattern and shift to αMβ2 integrin dependence for vascular monitoring[23,26,50]. MT4-MMP-mediated cleavage of αM integrin would likely destabilize the integrin heterodimer[51] allowing patrolling monocytes to detach once their surveillance function on the inflamed endothelium is performed. These results thus also position αMβ2 integrin as a major regulator of patrolling monocyte behavior in

AT, in which its role had remained unclear[52–54]. Interaction of αMβ2 integrin with ICAM-1, von Willebrand factor, fibrinogen, or CCN1 at athero-prone sites[55–58] might downregulate Foxp1 and induce c-fms[59], and hence influence MT4-MMP and Mafb expression (data not shown and[60]) as well as macrophage differentiation and/or proliferation.

This study has potential translational implications related to the atheroprotective action of MT4-MMP when present in BM-derived patrolling monocytes, with its expression increasing steadily with AT progression in mouse aorta and human coronary arteries. Genome-wide association studies will help to determine whether mutations in the MT4-MMP gene (MMP17) correlate with increased susceptibility to AT. Finally, patrolling monocytes have beneficial intravascular actions in infection, lung metastasis, and Alzheimer's disease[8,17,61], and our results suggest that short-term MT4-MMP targeting may offer a new therapeutic strategy to boost these activities.

## Methods

**Animal procedures**. MT4-MMP-deficient mice were generated as previously described[15]. Ldlr-deficient and Cx3cr1-Gfp/+ mice[62] were obtained from Jackson Laboratories. All mouse strains were on the C57BL/6 background. MT4-MMP−/−/Ldlr−/− and MT4-MMP−/−/Cx3cr1-Gfp/+ mice were generated by crossing MT4-MMP-deficient mice with Ldlr−/− or Cx3cr1-Gfp/+ mice. Mice used in this study included: $n = 58$ C57BL/6, $n = 67$ MT4-MMP−/−, $n = 154$ Ldlr−/−, $n = 20$ MT4-MMP−/−/Ldlr−/−, $n = 10$ Cx3cr1-Gfp/+, and $n = 10$ MT4-MMP−/−/Cx3cr1-Gfp/+. Mice were housed in the Centro Nacional de Investigaciones Cardiovasculares Carlos III (CNIC) Animal Facility under pathogen-free conditions and according to institutional guidelines. Animal studies were approved by the local ethics committee (CNIC Committee for Animal Welfare permit number CNIC-01/13 and local government of Madrid permit number PROEX 34/13) and conformed to directive 2010/63EU and recommendation 2007/526/EC regarding the protection of animals used for experimental and other scientific purposes, enforced in Spanish law under RD1201/2005. No statistical methods were used to pre-estimate the animal sample size and mice were randomly allocated to experimental groups. To induce atherosclerotic plaques, 3-month-old male mice were fed a HFD (ssniff, E15721-34) ad libitum for the indicated time. For BM transplantation, lethally irradiated (9 Gy) 8-week-old male Ldlr−/−MT4-MMP+/+ mice received tail vein injections of BM cells ($10^7$) obtained from tibias and femurs of euthanized donor Ldlr+/+MT4-MMP+/+ or Ldlr+/+MT4-MMP−/− mice. After 4 weeks on standard chow diet, transplanted mice were placed on the HFD for the indicated time. For CCR5 blocking, MRV (25 μg/g) or vehicle (5% dimethyl sulfoxide and 0.5% HCl 0.1 N in distilled water) were administered to BM transplanted Ldlr−/− mice by oral gavage every 12 h for 3 days, 7 days, or during the first week, and then every 24 h for a further 7 weeks until the end of the experiments. MRV was first administered at the end of the day previous to HFD feeding and last administered 2 h before sacrificing the mice. For adoptive transfer experiments, patrolling monocytes were sorted from the spleen of MT4-MMP+/+/Cx3cr1-Gfp/+ and MT4-MMP−/−/Cx3cr1-Gfp/+ mice as Lin (B220 (BD Pharmingen, 51.01122J), CD3 (BD Pharmingen, 51.01082J), Ly6G (BD Biosciences, 551461), and NK1.1 (BD Biosciences, 553165))-negative, GFP-positive and Ly6C-negative cells. Patrolling monocytes ($5 \times 10^5$) were injected intravenously into 3-month-old male Ldlr−/− mice; the mice were fed on the HFD for 3 days and then killed and the aortas processed for whole-mount staining. Investigators were not blinded during the analysis of mouse samples.

---

**Fig. 8** CCR5 inhibition results in loss of enhanced recruitment of patrolling monocytes, Mafb+AIM+ macrophage accumulation and AT acceleration in MT4-MMP-null BMT Ldlr−/− mice. **a** Scheme depicts the experimental design of CCR5 blocking strategy by Maraviroc (MRV) administration as described in M&M. **b** Representative orthogonal XZ view of confocal microscopy images of whole mount-stained aortic arch from Ldlr−/− mice transplanted with MT4-MMP+/+ or MT4-MMP−/− BM cells, fed the HFD for 3 days and treated with MRV or vehicle. Samples were stained for CD115 (red) and Ly6C (green); elastin autofluorescence (green) and nuclei (Hoechst, blue) are also shown. Scale bar, 20 μm. The bar graph (right) shows the quantification of the number of patrolling monocytes (CD115+Ly6C−) in the aorta lumen; $n = 7$ vehicle and $n = 8$ MRV mice per genotype in two independent experiments. **c** Bar graphs show the quantification of the number of Mac3+ cells (left), Mac3+Mafb+ cells (middle), and Mac3+Mafb+AIM+ cells (right) in the plaques of BM-transplanted Ldlr−/− mice fed a HFD for 1 week and treated with Maraviroc or vehicle. $n = 8$ mice per genotype and condition in two independent experiments. **d** Representative microscopy images of transverse sections of aortic sinus from BM-transplanted Ldlr−/− mice fed a HFD for 8 weeks and treated with Maraviroc or vehicle; sections were labeled for Mac3 (red) and Hoechst (blue; nuclei) (upper, scale bar, 50 μm) and graph shows the quantification of the number of Mac3+ cells in the plaque (lower). $n = 7$ vehicle and $n = 8$ MRV BM-transplanted mice per genotype in two independent experiments. **e** Graph showing the lesion area of aortas stained with Oil Red from BM-transplanted Ldlr−/− mice fed a HFD for 8 week and treated with MRV or vehicle. $n = 7$ and $n = 8$ mice for Vehicle and MRV mice per genotype in two independent experiments. **f** Stary scoring (I-VI) of transverse aortic H&E-stained sections from BM-transplanted Ldlr−/− mice fed a HFD for 8 weeks and treated with Maraviroc or vehicle, shown as a percentage of all mice for each score. $n = 7$ vehicle and $n = 8$ MRV BM-transplanted mice per genotype in two independent experiments. Data were tested by one-way ANOVA followed by Bonferroni's post test in **b–e** and by $\chi^2$-test for a trend in **f**. Results are expressed as mean ± SEM. $*p < 0.05$, $**p < 0.01$, $***p < 0.005$

**Human artery sampling and preservation.** Human coronary arteries ($n = 25$) were collected from freshly excised hearts during transplant operations at the Hospital de la Santa Creu i Sant Pau (HSCSP, Barcelona, Spain). The study was approved by the Hospital de la Santa Creu i Sant Pau Ethics Committee (04/2016) and was conducted according to the Declaration of Helsinki. Written informed consent was obtained from each patient. Immediately after surgical excision, arteries were dissected, cleaned of connective tissue, and examined under a dissecting microscope. Vessel samples were frozen in liquid nitrogen and stored at −80 °C for later protein or RNA extraction. Human samples were classified as early lesions (presenting only thickening of the intima) and established atherosclerotic lesions (presenting smooth muscle cells in the intima).

**Genotyping of BM-transplanted mice.** Blood samples were obtained from BM-transplanted mice 4 weeks after transplantation. White blood cells were isolated using Lympholyte (Cedarlane, CL5031) and DNA was extracted by isopropanol precipitation. PCR was performed using the following primers: forward WT: 5′-TCAGACACAGCCAGATCAGG-3′, forward KO 5′-AATATGCGAAGTG-GACCTGG-3′, and reverse (common to WT and KO) 5′-AGCAA-CACGGCATCCACTAC-3′. PCR was conducted at 94 °C for 2 min followed by 40 cycles of 95 °C for 40 s, 58 °C for 40 s, and 72 °C for 1 min, and a final elongation at 72 °C for 10 min.

**Biochemical analysis.** Total mouse serum cholesterol, triglycerides, and glucose were measured using the Dimensions RxL Max system (Siemens Healthineers).

**Atherosclerotic lesion analysis.** Hearts and aortas from euthanized mice were fixed with 4% of paraformaldehyde overnight at 4 °C. Adventitial fat and connective tissue were removed from aortas under a dissecting microscope. Whole aortas were opened longitudinally to expose the entire luminal surface and stained with 0.2% Oil Red O (Sigma-Aldrich, O0625) in 78% methanol. Images were acquired with a Nikon SMZ800 stereomicroscope (Nikon, Japan) coupled to a Nikon Coolpix 4500 digital color camera (Nikon). The Oil Red-positive area was measured using Image J (https://imagej.nih.gov/ij/; National Institutes of Health, Bethesda, MD).

**Histological and immunohistochemical analysis.** Mouse hearts and aortas were perfused with phosphate-buffered saline (PBS), extracted, fixed in 4% paraformaldehyde for 24 h at 4 °C, embedded in paraffin, and cut in 5 μm transverse sections for immunostaining or hematoxylin and eosin staining. Deparaffinized sections were rehydrated, and antigens were retrieved at 95 °C for 20 min in citrate buffer pH 6 or (for cleaved caspase-3 staining only) Tris-EDTA buffer pH 9. Sections were then left to cool to room temperature for 2 h. Antigen-retrieved paraffin sections and cryosections were blocked and permeabilized for 1 h at room temperature in PBS containing 0.3% (w/v) Triton X-100, 5% bovine serum albumin (BSA), 5% goat serum, and a 1:100 dilution of anti-CD16/CD32 (24g2, BD Pharmingen 553142). For immunofluorescence, sections were stained with anti-Mac3 (Santa Cruz Biotechnology, sc-19991), anti-MT4 (Abnova, PAB4785), anti-beta-galactosidase (rabbit, Abcam ab4761), anti-Ki67 (Abcam, ab16667), anti-cleaved caspase-3 (Cell Signaling Technology, 9661 S), anti-CD11c (eBiosciences, 11-014-81), anti-Mafb (Santa Cruz Biotechnology, sc-10022), anti-AIM (GeneTex, GTX37448), or anti-adipophilin (Novus Biologicals, NB110-40887) at 4 °C overnight or with anti-CD11b 647 (eBiosciences, 51-0112-82) for 2 h at room temperature. Primary antibodies were detected with corresponding fluorescent-labeled secondary antibodies. Samples were mounted in Fluoromount-G (SouthernBiotech, 0100-01) containing Hoechst 33342. Images were acquired on an inverted confocal microscope (LSM700, Carl Zeiss) fitted with a × 25 oil-immersion objective. Images were processed with Zen2009 Light Edition system (Carl Zeiss). Quantification was performed with Image J (https://imagej.nih.gov/ij/; National Institutes of Health) or CellProfiler software (Broad institute).

**Real time quantitative PCR.** Total RNA was isolated with TRIzol reagent (Invitrogen) using Maxtract high-density columns (Qiagen, ref. 1038988). Isolated RNA was treated with DNAse I (Sigma-Aldrich, D5025) and reverse transcribed using the High-Capacity cDNA Reverse Transcription Kit (Applied Biosystems). For mouse samples, quantitative PCR (qPCR) was performed with SYBR Green PCR master mix (Applied Biosystem) in a CFX384 detection system (Bio-Rad). Primers used in this study are listed in Supplementary Table 1. Data were normalized to the expression levels of 36b4 and cyclophilin mRNA within individual samples. For human samples, qPCR was performed using TaqMan probes for human *MMP17* (Hs00211754_m1) and normalized to the expression levels of TBP (Hs99999910_m1). All samples were analyzed in triplicate and RNA levels (CNRQ; calibrated normalized relative quantity) were calculated with Biogazelle qBase PLUS.

**Western blotting.** Protein extracts from human coronary artery samples were prepared in ice-cold lysis buffer containing 50 mM Tris-HCl pH 7.5, 1% (w/v) Triton X-100, 150 mM NaCl, and 1 mM dithiothreitol, and supplemented with protease inhibitors[63]. Mouse atheroma samples were obtained from paraffin-embedded samples, removing the paraffin, dewaxing, and rehydrating as follows: samples were placed in a tube and incubated for 30 min at 65 °C with 1 ml xylene and incubated at room temperature with rotation. Samples were then dehydrated through 100% ethanol (2 × 1 h), 96% ethanol (30 min), 70% ethanol (30 min), and PBS (2 × 20 min). Samples were lysed in 20 mM Tris-HCl, pH 7.5 containing 2% SDS and protease inhibitors for 20 min at 100 °C and 2 h at 60 °C. After centrifugation at 13,000 r.p.m. for 15 min at 4 °C, proteins from peritoneal macrophages were extracted in a buffer containing 10 mM Tris-HCl pH 7.5, 1% (w/v) Triton X-114, 150 mM NaCl, and protease inhibitors; samples were heated for 5 min at 30 °C to separate hydrophilic and lipophilic phases. Proteins were separated by 10% SDS-polyacrylamide gel electrophoresis, transferred to nitrocellulose membranes, and blocked with 5% non-fat milk. Primary antibodies used were anti-MT4-MMP (Abcam, ab51075), anti-CD11b (Abcam, ab75476), anti-caveolin1 (Cell Signaling, 3267 S), anti-α-actin (Dako, M0851), and anti-tubulin (Sigma-Aldrich, T6074). After overnight incubation at 4 °C and washes, bound primary antibody was detected by incubation for 1 h with donkey-anti-rabbit (IRDyeTM 800CW or 680CW, 1/10,000, Odyssey) or donkey-anti-mouse (IRDyeTM 800CW or 680CW, 1/10,000, Odyssey) secondary antibodies, followed by visualization with the Odyssey Infrared Imaging System (LI-COR Biosciences). Raw western blotting images are presented in Supplementary Fig. 12.

**In silico modeling of MT4-MMP dimer/Itgam-Itgb2 integrin heterodimer.** Fasta sequences of mature human MT4-MMP/MMP17, Itgam, and Itgb2 proteins were analyzed with locally implemented I-Tasser suite v4.4 for threading modeling[64]. Selected models were those with minimal energy and correct folding (best structural alignment to templates). In these models, the TM domain is buried in the protein core. Therefore, angles in this region (TM plus C-term) were fixed according to the predicted secondary structure and a refinement cycle was performed with the membrane framework of Rosetta suite v3.5 release 2015.38.58158 (www.rosettacommons.org)[65]. To dock dimeric MT4-MMP to Itgam-Itgb2, the monomers modeled previously were positioned according to the published dimeric interface using pymol v1.8 (www.pymol.org) and the new dimeric model was used as the initial template. Using this model as input, with the same spanfile and constraints as before, a new dimer model was generated using the *mp_dock* application from the membrane framework of Rosetta suite v3.5 release 2015.38.58158 in each case[65]. Using the obtained dimer models, the MT4-MMP-Itgam-Itb2 complex was modeled by a similar approach.

**Identification of cleavage sites and in vitro digestion assay.** Cleavage sites for MT4-MMP in αMβ2 integrin were identified by Cleavpredict[22] (http://cleavpredict.sanfordburnham.org/) and the predicted and exposed sites were then filtered according to the peptide cleavage matrix in the MEROPS database (http://merops.sanger.ac.uk/). Conservation of cleavage sites in the mouse and human αM and αL integrin chains was checked in Uniprot (http://www.uniprot.org/align/). The αM integrin peptide (RPQVTFSENLSSTCHTKER) was synthetized at the Centro de Investigaciones Biológicas (CIB, CSIC, Madrid) and incubated with hrMT4-MMP (RP-77535, Thermo Fisher Scientific) for 2 h at 37 °C in water. The resulting peptides were assayed by high-resolution parallel reaction monitoring on an Easy nLC 1000 nano-HPLC apparatus (Thermo Fisher Scientific) coupled to a hybrid quadrupole-orbitrap mass spectrometer (Q Exactive, Thermo Scientific). The peptides were separated at 200 nL m$^{-1}$ in a continuous gradient consisting of 8–30% B for 15 min and 30–90% B for 2 min ($B$ = 90% acetonitrile, 0.1% formic acid) and ionized using a Picotip emitter nanospray needle (New Objective, Woburn, MA, USA). Each mass spectrometric (MS) run consisted of enhanced FT-resolution spectra (15,000 resolution) in the 400–1400 m z$^{-1}$ range followed by data-independent tandem MS spectra of seven parent ions acquired during the chromatographic run. The AGC target value in the Orbitrap for the survey scan was set to 1,500,000 and fragmentation was performed at 27% normalized collision energy with a target value of 250,000 ions. Data were analyzed with Xcalibur 2.2 (Thermo Fisher Scientific).

**Peritoneal macrophage analysis.** Primary mouse peritoneal macrophages were obtained from MT4-MMP$^{+/+}$ or MT4-MMP$^{-/-}$ mice by i.p. lavage with 10 ml cold PBS 72 h after i.p. injection of 3% (w/v) TG. Macrophages were counted, pelleted, and resuspended in RPMI-1640 (Sigma-Aldrich) supplemented with 2.5% fetal bovine serum, 10 mM HEPES, 50 UI ml$^{-1}$ penicillin, 50 μg ml$^{-1}$ streptomycin, 1 mM sodium pyruvate, and 0.1 mM non-essential amino acids, and plated on coverslips for spreading experiments or on plastic tissue culture dishes for mRNA or protein analyses. Freshly isolated peritoneal cells were used for flow cytometry. For MafB and AIM immunofluorescence analysis of TG-elicited peritoneal macrophages, cells were plated on glass coverslips (500,000 cells per 12 mm-diameter coverslip) for 24 h, fixed with 4% paraformaldehyde for 10 min at room temperature, and blocked and permeabilized for 30 min at room temperature in PBS containing 0.3% (w/v) Triton X-100, 5% BSA, 5% goat serum, and a 1:100 dilution of anti-CD16/CD32 (24g2, BD Pharmingen 553142). Fixed cells were stained with anti-Mafb (Santa Cruz Biotechnology, sc-10022) and anti-AIM (GeneTex, GTX37448) overnight at 4 °C, and with anti-CD11b 647 (eBiosciences, 51-0112-82) for 2 h at room temperature, followed by the corresponding fluorescent-labeled secondary antibodies. Apoptosis was assessed by treating TG-elicited peritoneal

macrophages with cycloheximide (100 µg ml$^{-1}$) for 6 h and staining fixed cells with anti-cleaved caspase 3 (Cell Signaling, 9661 S) as described above. Samples were mounted in Fluoromount-G (SouthernBiotech, 0100-01) containing Hoechst 33342. Images were acquired on an inverted confocal microscope (LSM700, Carl Zeiss) fitted with a × 25 oil-immersion objective. Images were processed and analyzed using Image J (https://imagej.nih.gov/ij/; National Institutes of Health). For C3 and acLDL binding assays, TG-elicited peritoneal macrophages were plated over coverslips or left in suspension and incubated with C3-opsonized sheep red blood cells (RBCs) or with 1 µg ml$^{-1}$ acLDL-FITC (Invitrogen, L-23380) for the indicated times at 37 °C; the number of RBC per macrophage was then counted and the fluorescence intensity of AcLDL-FITC quantitated by flow cytometry. Macrophage egression was assessed as described[52]. In brief, TG-elicited peritoneal macrophages from MT4-MMP$^{+/+}$ or MT4-MMP$^{-/-}$ mice were labeled with PKH26 or PKH67 probes (Sigma-Aldrich MINI26 and MINI67), respectively, and mixed 1:1. Dual-labeled macrophages (5 × 10$^5$) were injected i.p. into C57BL6 mice previously stimulated with TG i.p. for 72 h. LPS (1 µg) or PBS was then injected i.p. and after 4 h mice were killed and peritoneal macrophages and spleen obtained for flow cytometry analysis.

**In vivo macrophage infection**. Lentiviruses encoding full-length mouse MT4-MMP, MT4-MMP E248A (catalytic inactive mutant), or GFP alone (mock) were generated as described[16]. Virus was i.p. injected into MT4-MMP$^{+/+}$ or MT4-MMP$^{-/-}$ mice at ~ 1 × 10$^8$ pfu ml$^{-1}$. After 5 days, cells were collected and analyzed by flow cytometry.

**Flow cytometry**. Blood and BM samples from killed mice were blocked for 15 min at 4 °C in PBS containing 5% BSA and a 1:100 dilution of anti-CD16/CD32 (24g2, BD Pharmingen, 553142). Samples were then stained for 30 min at 4 °C with anti-CD11b 647 (eBiosciences, 51-0112-82) or biotinylated (BD Pharmingen, 51.01712 J), anti-CD45 V450 (eBiosciences, 48-0451-82), anti-Ly6C FITC (BD Bioscience, 553104) or APC (BD Pharmingen, 560595), anti-CCR5 PE (eBioscience, 12.1951-12), and anti-CCR2 (Biolegend, 150607) primary antibodies and with streptavidin PE (BD Bioscience, 554061) secondary reagent. Erythrocytes in blood samples were lysed with FACS Lysis Solution (BD Biosciences, 349202) for 7 min at room temperature. Before gating, granulocytes were excluded by FCS/SSC. Peritoneal macrophages were collected on the indicated days after TG administration and blocked with BSA/anti-CD16/CD32. Macrophages were then stained for 30 min at 4 °C with anti-CD11b 647 (eBiosciences, 51-0112-82), anti-CD45 V450 (eBiosciences, 48-0451-82), anti-F4/80 Pe-Cy7 (Biolegend, 123114), anti-AIM (GeneTex, GTX37448), and anti-CD36 (Cascade BioScience, ABM-5525) antibodies; for quantification of dead cells, Hoechst 33258 (Sigma, 861405) was added 5 min previous to flow cytometry analysis. Data were acquired in a FACSCanto III cytometer (BD) and analyzed using FlowJo software (Tree Star).

**Cell spreading assay**. Peritoneal macrophages were plated onto fibrinogen-coated coverslips (500,000 cells per 12 mm-diameter coverslip) for 24 h, fixed with 4% paraformaldehyde for 10 min at room temperature, stained with phalloidin-texas red (Invitrogen, T-7471, 1:100) for 2 h, and mounted in Fluoromount-G (SouthernBiotech, 0100-01) containing Hoechst 33342. Images were acquired with a confocal microscope (Nikon A1R) fitted with an × 20 air objective. Images were processed and analyzed using Image J (https://imagej.nih.gov/ij/; National Institutes of Health).

**Whole-mount staining of peritoneal membranes**. Peritoneal membranes were collected 72 h after TG administration, cleaned under a dissecting microscope, and fixed with 4% paraformaldehyde overnight at 4 °C. Samples were blocked and permeabilized for 30 min at room temperature in PBS containing 0.1% Triton X-100, 5% BSA, 5% goat serum (Jackson, 005-000-001), and 1:100 anti-CD16/CD32 (24g2, BD Pharmingen, 553142). Samples were then incubated with anti-CD11b 647 (eBiosciences, 51-0112-82) overnight at 4 °C and mounted in Fluoromount-G (SouthernBiotech, 0100-01) containing Hoechst 33342. Images were acquired with an inverted confocal microscope (LSM700, Carl Zeiss) fitted with a × 10 air objective. Images were processed with Zen2009 Light Edition (Carl Zeiss), and quantified using Image J (https://imagej.nih.gov/ij/; National Institutes of Health).

**Intravital microscopy in the cremaster muscle**. Intravital microscopy in the cremaster muscle was performed as described[25]. Mice were anesthetized and the cremaster muscle was dissected free of surrounding tissues and exteriorized onto an optical clear viewing pedestal. The muscle was cut longitudinally with a cautery and held extended at the corners of the exposed tissue using surgical suture. To maintain the correct temperature and physiological conditions, the muscle was perfused continuously with warmed Tyrode's buffer. Four hours before surgery, animals were injected intrascrotally with 345 ng of CCL2 (Preprotech, 250–10)[66]. The cremasteric microcirculation was then observed using a Leica DM6000-FS intravital microscope fitted with an Apo × 40 NA 1.0 water-immersion objective and a DFC350-FX camera. LASAF software was employed for acquisition and image processing. Monocytes were stained by i.v. injection of CD115-PE (Biolegend, 135505) and Ly6C-APC (Biolegend, 128016), and in another set of experiments neutrophils were stained by injection of Ly6G-PE (BD Biosciences, 551461).

Three-to-five randomly selected venules (25–40 µm diameter) were analyzed per mouse, and leukocyte-endothelium interaction was measured in 350 µm vessel segments for 5 min. When indicated, the αM integrin (Itgam) blocking antibody M1/70 (eBioscience, 16-0112) or IgG isotype control (eBioscience, 16-4031) was injected intravenously (4 mg kg$^{-1}$) before intravital microscopy[26].

**Statistical analysis**. All data are shown as mean ± SEM. Normal distribution of the values was checked and statistical analysis performed with Prism Software (GraphPad Prism 5) using the test indicated in the figure legend. Outlier values were excluded using the online GraphPad outlier test. Statistical significance was assigned at *$p < 0.05$, **$p < 0.01$, and ***$p < 0.001$.

**Data availability**. All relevant data are available from the authors upon request.

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

## Acknowledgements

We thank Ángel Colmenar and Laura Balonga for technical support, the BioBanco VIH (Hospital Gregorio Marañón, Madrid) for providing essential reagents, and Simon Bartlett for English editing. This study was supported by grants from the Spanish Ministry of Economy, Industry and Competitiveness (MEIC; RD12/0042/0023 [FEDER cofunded] and SAF2014-52050R and SAF2017-83229R to A.G.A., SAF2015-64287R and SAF2015-71878-REDT to M.R., RD12/0042/0053 [FEDER cofunded] and SAF2015-64767-R to J.M-G., and SAF2016-79490-R and RD12/0042/0028 [FEDER cofunded] to V.A.) and from La Marató TV3 Foundation. C.C., M.M-A., and L.A-H. were funded by fellowships from the Spanish Ministry of Education, MEIC, and La Caixa-CNIC, respectively. C.R. was funded by a competitive postdoctoral contract grant FPDI-2013-17423 from MEIC. The CNIC is supported by the Spanish MEIC and the Pro-CNIC Foundation, and is a Severo Ochoa Center of Excellence (MEIC award SEV-2015-0505).

## Author contributions

C.C. performed and analyzed most of AT and macrophage experiments. A.P. and R.A. M. performed and analyzed some AT experiments in *Ldlr*−/− and BM-transplanted *Ldlr*−/− mice. V.N. helped with BM transplants and Oil Red aorta analysis. C.R. performed and analyzed intravital microscopy experiments. L.A-H. performed and analyzed qPCR. M.M.-A. helped in digestion assays. E.C. performed and analyzed mass spectrometry experiments. F.M. performed in silico protein modeling. C.R. and J. M-G. analyzed human samples. M.S. provided MT4-MMP-deficient mice. V.A. supervised

intravital microscopy assays. M.R. provided $Ldlr^{-/-}$ and Cx3cr1$^{Gfp/+}$ mice, and critical suggestions. A.G.A. designed and supervised the research and wrote the paper.

## Additional information

**Competing interests:** The authors declare no competing interests.

