## [Peer Review File · Nature Communications]

Reviewers' comments:

Reviewer #1 (Remarks to the Author):

The authors present an interesting set of studies that highlight the role of MT4-MMP in atherogenesis, particularly in regard to the contribution of the Ly6Clow subset of monocytes, also referred to as "patrolling" monocytes. The data are well presented and appear to be generally sound, but the following issues/comments should be considered:

1. One issue is that the authors did not consider the possibility that the increased adherence phenotype could also lead to increased retention of the mutant monocyte derived macrophages in the plaque leading to their phenotype. They do not attempt to rule out lack of egress as a factor in their atherosclerosis phenotype observed in their MT4-MMP null/Ldlr-/- BMT mice. Indeed, of the kinetic regulators of plaque macrophage content (recruitment, apoptosis, proliferation, and egress), this is the only one not studied.
2. Atherosclerosis progression is classically characterized by Ly6Chigh monocyte recruitment. Ly6Chigh monocytes use CCR2 and CX3CR1 to enter plaques while Ly6Clow monocytes use CCR5 (Tacke et al, 2007, JCI, as the authors cite). Combadiere et al 2008 (Circulation) showed a 75% decrease in macrophage content in atherosclerotic plaques when CCR2 and CX3CR1 are both knocked out and 90% when CCR2, CX3CR1 and CCR5 are all inhibited, suggesting Ly6Chigh monocytes are the major source of macrophages in the lesions. In this study, the null mice have double the amount of macrophages at 8 weeks on diet. If all of this was due to the low subset, the quantitative impact is surprisingly large. Have the authors tested the possibility that MT4-MMP may also effect Ly6Chigh recruitment and adhesion to inflamed endothelium? One useful experiment would be to treat the mice with a CCR5 inhibitor to see if you lose the increase in macrophage content (which would suggest the phenotype is specific to Ly6Clow patrolling monocytes. Do MT4-MMP null macrophages have higher CCR5 expression suggesting they use CCR5 to enter inflamed areas (but no change in CCR2 or CX3CR1)?
3. In the abstract the authors mention that the thioglycolate elicited MT4-MMP null macrophages expressed higher surface levels of AIM and were more resistant to apoptosis (Figure 6). They also see higher AIM expression on macrophages in MT4-MMP null/Ldlr-/- mice plaques after 7 days HFD (Figure 5). However they also observe larger necrotic core area in these mice at 12 weeks HFD (Figure 2). A main part of necrotic core formation is macrophage death via apoptosis. At the 8 and 12 week HFD time point, do the MT4-MMP null macrophages have decreased AIM expression than controls? If they have increased AIM expression, what mechanism is behind the larger necrotic core area?
4. The authors mention that patrolling monocytes have beneficial effects in infections and the prevention of lung metastasis and therefore boosting their function through inhibition of MT4-MMP could be therapeutically beneficial. However, wouldn't this also increase atherosclerosis progression?
5. It is not clear the relationship between MT4-MMP status and the expression of the transcription factor Mafk. Is there a causal relationship, or simply that they are both expressed by a particular subset whose abundance is regulated by other factors?
6. Supplemental Fig. 2: something is wrong with the glucose measurements; not even diabetic mice go that high.
7. It is not clear to me why the HFD has such rapid effects. In studies of atherosclerosis progression, it takes at least 2 weeks to stably increase cholesterol levels, and about 8 weeks to see sub-endothelial accumulation of macrophages. How do the changes begin within a week? Also, in Supp. Figure 4, the graph indicates 3 days of feeding the HFD, but the legend and text say 1

week.

8. The authors write: "These data pointed to enhanced monocyte recruitment during early atherosclerosis as the main contributor to increased macrophage burden". However, the if expression of MT4-MMP is very low (barely detectable) at 8 weeks- if it is so low, why would it impact early lesions so much?

Reviewer #2 (Remarks to the Author):

The authors demonstrate that MT4-MMP deficiency is associated with increased macrophage adherence to the inflamed peritoneum that appears to be mediated by Integrin alpha M (ITGAM), which is a substrate for MT4-MMP. It is proposed that MT4-MMP cleavage of ITGAM mediates endothelial patrolling of Ly6Clo monocytes. In a separate line of inquiry, the authors demonstrate that bone marrow-derived MT4-MMP-deficiency also contributes to increased atherosclerosis burden. It is hypothesized that this is due to the increased recruitment of Ly6Clo monocytes that differentiate to atherogenic MafB+ macrophages.

Characterization of increased atherosclerosis in MT4-MMP-deficient animals is convincing. However, there is no direct evidence linking either increased Ly6Clo monocyte recruitment or MafB+ macrophages to the observed phenotype. The studies carried out in the cre-master muscle-based system can NOT, as the authors suggest, be directly translated to the in vivo plaque environment. Furthermore, the adoptive transfer studies reported in Figure 4 are confusing. Transfer of whole bone marrow does not allow one to directly assess trafficking of Ly6Clo monocytes.

Figure 5 shows that MT4-MMP-deficiency associates with increased abundance of MafB+ macrophages. Is there any direct evidence that MafB+ macrophages represent a functionally distinct macrophage subset that uniquely contributes to atherosclerosis severity? Given the total number of macrophages is similar between wild type and MT4-MMP-deficient mice, this needs to be tested. Overall, it is the feeling of this reviewer that observations in unrelated experimental systems are over-interpreted as playing a role in the development of atherosclerosis.

POINT-BY-POINT REPLY

Reviewer #1 (Remarks to the Author):

The authors present **an interesting set of studies** that highlight the role of MT4-MMP in atherogenesis, particularly in regard to the contribution of the Ly6Clow subset of monocytes, also referred to as “patrolling” monocytes. The **data are well presented and appear to be generally sound**, but the following issues/comments should be considered:

1. One issue is that the authors did not consider the possibility that the increased adherence phenotype could also lead to increased retention of the mutant monocyte derived macrophages in the plaque leading to their phenotype. They do not attempt to rule out lack of egress as a factor in their atherosclerosis phenotype observed in their MT4-MMP null/Ldlr-/- BMT mice. Indeed, of the kinetic regulators of plaque macrophage content (recruitment, apoptosis, proliferation, and egress), this is the only one not studied.

We concur with the reviewer that we cannot rule out that macrophage egression may have a role in the observed phenotype, although recent studies have questioned the actual contribution of egression to the overall macrophage burden not only in atherosclerotic plaques but also in other inflammatory contexts (Gautier et al., 2013; Randolph, 2015 and references herein). Nevertheless, and though local death seems to be the key mechanisms for decreasing macrophage abundance, we will check macrophage egression to lymph nodes in the TG-peritonitis model by dual-labeling WT and MT4-MMP-null peritoneal macrophages with fluorescent probes currently available in our laboratory (Gautier et al., 2013; Cao et al., 2005).

2. Atherosclerosis progression is classically characterized by Ly6Chigh monocyte recruitment. Ly6Chigh monocytes use CCR2 and CX3CR1 to enter plaques while Ly6Clow monocytes use CCR5 (Tacke et al, 2007, JCI, as the authors cite). Combadiere et al 2008 (Circulation) showed a 75% decrease in macrophage content in atherosclerotic plaques when CCR2 and CX3CR1 are both knocked out and 90% when CCR2, CX3CR1 and CCR5 are all inhibited, suggesting Ly6Chigh monocytes are the major source of macrophages in the lesions. In this study, the null mice have double the amount of macrophages at 8 weeks on diet. If all of this was due to the low subset, the quantitative impact is surprisingly large. Have the authors tested the possibility that MT4-MMP may also effect Ly6Chigh recruitment and adhesion to inflamed endothelium? One useful experiment would be to treat the mice with a CCR5 inhibitor to see if you lose the increase in macrophage content (which would suggest the phenotype is specific to Ly6Clow patrolling monocytes. Do MT4-MMP null macrophages have higher CCR5 expression suggesting they use CCR5 to enter inflamed areas (but no change in CCR2 or CX3CR1)?

We have already analyzed the possible effect of MT4-MMP on the Ly6C^{high} population as reviewer 1 requested. We found no differences in total monocyte rolling and adherence to the inflamed endothelium of the cremaster muscle or in Ly6C^{high} monocyte adherence to the inflamed aorta 3 days after HFD in the absence of MT4-MMP (**rebuttal Figure 1**). These new data have been included in the revised manuscript (lines 206-209 and lines 219 and 221 and **new Figure 4c and Figure 5**, bottom) and strongly suggest that MT4-MMP deficiency mainly influences patrolling monocyte behavior.

Rebuttal Figure 1. Adherence of Ly6Chigh classical monocytes to the inflamed endothelium in the absence of MT4-MMP. A, Bar graph show the quantification of the number of monocytes (CD115+Ly6G-) rolling (left) and adhered (right) to CCL2-stimulated endothelium in the cremaster muscle analyzed by intravital microscopy. n=8 mice per genotype. B, Bar graph shows the quantification of the number of CD115+Ly6C+ monocytes adhered to the lumen of aortas from Ldlr-/- mice transplanted with MT4+/+ or MT4-/- bone marrow cells and fed a HFD during 3 days. n=6 mice per genotype.

We are also extremely grateful to reviewer 1 by the suggestion of inhibiting CCR5 receptor in vivo. We already have the CCR5 inhibitor Maraviroc available (provided by our collaborator Dr. MA Muñoz, Hospital Gregorio Marañón, Madrid) and we are launching this experiment which would demonstrate the causal link between enhanced patrolling monocyte adherence and the macrophage atherosclerotic phenotype observed in the absence of MT4-MMP.

Moreover, as proposed by the reviewer, we have checked the expression of CCR5, CCR2 and CX3CR1 in MT4-MMP-null macrophages by qPCR (**rebuttal Figure 2**) and found no differences compared to wild-types. These data further support that MT4-MMP absence mainly impacts on the levels and activity of the α M β 2 integrin rather than on chemokine receptor regulation.

Rebuttal Figure 2. Chemokine receptor expression in MT4-MMP-null peritoneal macrophages. Bar graph shows mRNA relative levels of the chemokine receptors Ccr5, Ccr2 and Cx3cr1 and of Mmp17 quantitated by qPCR in peritoneal macrophages from wild-type (MT4^{+/+}) and MT4-MMP-null (MT4^{-/-}) mice obtained 72 h after TG stimulation and adhered to plastic dishes overnight. n=6 mice per genotype. Data were analyzed by Student's t-test.

3. In the abstract the authors mention that the thioglycolate elicited MT4-MMP null macrophages expressed higher surface levels of AIM and were more resistant to apoptosis (Figure 6). They also see higher AIM expression on macrophages in MT4-MMP null/Ldlr^{-/-} mice plaques after 7 days HFD (Figure 5). However they also observe larger necrotic core area in these mice at 12 weeks HFD (Figure 2). A main part of necrotic core formation is macrophage death via apoptosis. At the 8 and 12 week HFD time point, do the MT4-MMP null macrophages have decreased AIM expression than controls? If they have increased AIM expression, what mechanism is behind the larger necrotic core area?

As requested by the reviewer, we have quantitated AIM-positive macrophages in BMT-Ldlr^{-/-} mice fed a HFD for 8 and 12 weeks (**rebuttal Figure 3**); these data have been included in the revised manuscript (lines 242-246 and **new Supplementary Figure 7a and 7b**). Notably, and as predicted by the reviewer, the increased number and % of Mac3+AIM+ macrophages present after 1 week HFD in the MT4-MMP-null BMT Ldlr^{-/-} mice compared to those transplanted with wild-type cells remained up to 8 weeks but was no longer observed after 12 weeks HFD. As pointed out by the reviewer, these data may explain the trend to larger necrotic area observed in MT4-MMP-null BMT Ldlr^{-/-} mice after 12 weeks HFD.

Rebuttal Figure 3. AIM expression in macrophages in advanced AT plaques in the absence of MT4-MMP. A, Representative confocal microscopy images of transverse sections from the aortic sinus of *Ldlr*^{-/-} mice transplanted with MT4^{+/+} or MT4^{-/-} bone marrow cells and fed a HFD for 8 or 12 weeks. Mac-3 (green), AIM (red), Hoechst (blue). Scale bar correspond to 50 μ m. L indicate the lumen. B, Bar graph shows the quantification of the number of double Mac-3/AIM positive macrophages (left) and the percentage of AIM⁺ macrophages (right). n=5 mice per genotype and time point.

4. The authors mention that patrolling monocytes have beneficial effects in infections and the prevention of lung metastasis and therefore boosting their function through inhibition of MT4-MMP could be therapeutically beneficial. However, wouldn't this also increase atherosclerosis progression?

Our data identify MT4-MMP targeting as the first molecular intervention with potential to increase patrolling monocyte activity. If MT4-MMP targeting eventually make it to the clinics, it will be a short-term and likely adjuvant therapy to treat acute infections or prevent lung metastasis. This is not expected to influence overall natural history of atherosclerosis given that this is a torpid disease taking years of evolution in the human patient (not just months as in the mouse model). This point has been smoothed in the corresponding line of the discussion in the revised manuscript (line 367).

5. It is not clear the relationship between MT4-MMP status and the expression of the transcription factor Mafb. Is there a causal relationship, or simply that they are both expressed by a particular subset whose abundance is regulated by other factors?

The literature and our own data support that MT4-MMP and Mafb can be expressed in particular macrophage subsets and regulated by not-yet and maybe shared mechanisms. As an example, MT4-MMP and Mafb are both upregulated during M-CSF-driven differentiation of bone marrow-derived macrophages (**rebuttal Figure 4**; Cuevas et al., 2017 Figure 1B). This point could be mentioned in the discussion if required.

Rebuttal Figure 4. MT4-MMP expression in M-CSF bone marrow-derived macrophages. Representative western-blot of MT4-MMP expression in cell lysates from mouse bone marrow-derived macrophages (after 7 days treatment with M-CSF) obtained from wild-type (MT4-MMP+) and MT4-MMP-null (MT4-MMP-) mice. β -actin is included as loading control.

In relation to their possible inter-regulation, MT4-MMP (MMP17) does not seem to be a direct Mafb target gene since it does not appear among genes described as regulated by Mafb (Cuevas et al., 2017). On the other hand, our data show that MT4-MMP absence influences the behavior of Mafb+ macrophages rather than Mafb expression itself (no changes in Mafb mRNA levels are detected by qPCR in MT4-MMP-null TG-peritoneal macrophages, revised Supp Figure 8b). This can be discussed further in the revised version.

6. Supplemental Fig. 2: something is wrong with the glucose measurements; not even diabetic mice go that high.

We regret this mistake due to the involuntary use of the wrong graphic scale for the glucose levels and we have amended this point in the revised version (Annex I and revised Supplementary Fig. 2c)).

7. It is not clear to me why the HFD has such rapid effects. In studies of atherosclerosis progression, it takes at least 2 weeks to stably increase cholesterol levels, and about 8 weeks to see sub-endothelial accumulation of macrophages. How do the changes begin within a week? Also, in Supp. Figure 4, the graph indicates 3 days of feeding the HFD, but the legend and text say 1 week.

Our data are in line with previous studies showing lipid deposits in athero-prone aortic regions as early as 5 days after HFD feeding (Zhou et al., 2009; Paulson et al., 2010; Randolph, 2015 and references herein) even in the absence of detectable plasma cholesterol changes. Confocal microscopy analysis allows the detection of the few lipid-loaded macrophages present in the endothelial/sub-endothelial area at this early stage and that are not-yet visible by regular Red Oil staining or H&E histology.

We think the reviewer refers to main Figure 4 (instead of Supp. Figure 4); we regret the mistake in the time-point mentioned in the legend and we have amended it in the legend of the revised version (revised Figure 5).

8. The authors write: "These data pointed to enhanced monocyte recruitment during early atherosclerosis as the main contributor to increased macrophage burden". However, the if expression of MT4-MMP is very low (barely detectable) at 8 weeks- if it is so low, why would it impact early lesions so much?

We appreciate the reviewer's doubt about MT4-MMP expression data during atherosclerosis progression and the observed phenotype and we think that it may be related to how these data were originally presented. On one hand, as pointed out by the reviewer, MT4-MMP protein and mRNA expression is low in early mouse

atherosclerotic plaques (8 weeks HFD) but it is important to notice that this expression is in total aorta tissue extracts in which MT4-MMP macrophage contribution will be diluted. Nevertheless in these aortic plaques at 8 weeks, MT4-MMP is mostly expressed in macrophages. Moreover, the atherosclerotic phenotype in the absence of MT4-MMP is observed as early as 1 week after HFD (Figures 5 and 6), and as shown in the new Supp Figure 2d MT4-MMP is already present in macrophages in these incipient lesions in which we detect the first alterations in macrophage composition. We have included a new panel with MT4-MMP expression at 1 week, re-ordered the contents, and re-written this paragraph for further clarity (lines 127-130 and lines 152-155 and revised **Supplementary Figures 2d and 4**).

Reviewer #2 (Remarks to the Author):

The authors demonstrate that MT4-MMP deficiency is associated with increased macrophage adherence to the inflamed peritoneum that appears to be mediated by Integrin alpha M (ITGAM), which is a substrate for MT4-MMP. It is proposed that MT4-MMP cleavage of ITGAM mediates endothelial patrolling of Ly6Clow monocytes. In a separate line of inquiry, the authors demonstrate that bone marrow-derived MT4-MMP-deficiency also contributes to increased atherosclerosis burden. It is hypothesized that this is due to the increased recruitment of Ly6Clow monocytes that differentiate to atherogenic Mafb+ macrophages.

Characterization of increased atherosclerosis in MT4-MMP-deficient animals is convincing. However, there is no direct evidence linking either increased Ly6Clow monocyte recruitment or MafB+ macrophages to the observed phenotype.

We appreciate the positive comment of the reviewer 2 on the characterization of the atherosclerosis phenotype. CCR5 inhibition strategy (as proposed by reviewer 1) will provide the direct evidence between increased Ly6Clow monocyte recruitment and Mafb+ macrophage accumulation to the observed atherosclerotic phenotype in the absence of MT4-MMP. These experiments are currently being launched.

*The studies carried out in the cre-master muscle-based system can NOT, as the authors suggest, be directly translated to the in vivo plaque environment. Furthermore, **the adoptive transfer studies reported in Figure 4 are confusing.** Transfer of whole bone marrow does not allow one to directly assess trafficking of Ly6Clow monocytes.*

Although CCL2 is a well-recognized player in the initiation of atherosclerosis (Charo and Taubman, 2004), we agree that the cremaster muscle environment cannot fully recapitulate incipient atherosclerotic lesions in the aorta and we have smoothed this sentence in the revised version (lines 200-201).

We are however particularly concerned by the reviewer's comment '**Furthermore, the adoptive transfer studies reported in Figure 4 are confusing**' since no adoptive transfer studies are reported whatsoever in Figure 4 not in any other of the study. This comment may reflect a superficial reading/misunderstanding of the findings that we hope has not influenced his/her final perception and evaluation of our manuscript. We have analysed the traffic of Ly6Clow monocytes in MT4-MMP-deficient mice (Figure 4a and b) and in MT4-MMP-null BM transplanted Ldlr-/- mice (revised Figure 5) by specific marker labelling. Complementary approaches (in reporter mice or in mice lacking patrolling monocytes) could be performed if required.

Figure 5 shows that MT4-MMP-deficiency associates with increased abundance of MafB+ macrophages. Is there any direct evidence that MafB+ macrophages represent a functionally distinct macrophage subset that uniquely contributes to atherosclerosis severity? Given the total number of macrophages is similar between wild type and MT4-MMP-deficient mice, this needs to be tested.

Takahashi's group showed the presence of Mafb in a subset of macrophages in the atherosclerotic plaque and established the first evidence that Mafb participated in the acceleration of atherogenesis by regulating AIM (apoptosis inhibitor of macrophages) (Hamada et al., 2014). Our study provides additional data that support that the Mafb+ macrophage subset can also express CD36 and thus uptake modified LDL. In spite of similar number of macrophages between-genotypes at 7 days after HFD, in the absence of MT4-MMP the increased percentage of Mafb+ macrophages expressing higher levels of AIM will make them more prone to survive and to accumulate lipids than Mafb-AIM- macrophages leading to the observed increase in macrophage number and in lipid deposits in the aortas of MT4^{-/-}-transplanted Ldlr-/- mice at 8 and 12 weeks after HFD, respectively. We have also observed that in incipient lesions, Mafb+ macrophages proliferate more in the absence of MT4-MMP (**rebuttal Figure 5**); this information has been included in the text of the revised manuscript (lines 236-238). Therefore in incipient lesions in the absence of MT4-MMP the macrophage Mafb+ subset will have enhanced anti-apoptotic and proliferative capacities and will uptake lipids what would favor their accumulation in advanced lesions and therefore the acceleration of atherosclerosis.

Rebuttal Figure 5. Mafb⁺ macrophages are more proliferative in the absence of MT4-MMP in incipient atherosclerotic lesions. Representative confocal microscopy images of transverse sections from the aortic sinus from *Ldlr*^{-/-} mice transplanted with MT4^{+/+} or MT4^{-/-} bone marrow cells and fed a HFD during 1 week (left panel). Mac3 (green), Mafb (red), Ki67 (grey) and nuclei (blue). Scale bar correspond to 20µm. L indicate the lumen. Bar graph (right panel) shows the percentage of Mac3+Mafb+ cells that are or not proliferating (Ki67+ vs Ki67-). n=7 mice per genotype. Data were tested with Fisher's exact test. *p<0,05.

Overall, it is the feeling of this reviewer that observations in unrelated experimental systems are over-interpreted as playing a role in the development of atherosclerosis.

The three models used in our work, thioglycollate-peritonitis, CCL2-stimulated endothelium in cremaster muscle and atherosclerosis are not unrelated experimental systems but complementary in vivo mouse models of inflammation, all of them involving activation of the endothelium and recruitment of monocytes which will eventually differentiate into macrophages. We hope this reviewer's statement is not related to misreading/misinterpretation of part of the data (i.e. Figure 4) and we are sure the new data (particularly CCR5-inhibition strategy) will strengthen the conclusions of our study about the involvement of MT4-MMP-null patrolling monocyte-derived macrophages in atherosclerosis development.

ANNEX I

Supplementary Fig. 2. AT-related parameters in *Ldlr*-deficient mice transplanted with MT4-MMP-null BM cells. (a) Engraftment analyzed by PCR of genomic DNA from blood cells of BM-transplanted *Ldlr*^{-/-} mice. (b) Body weight and (c) serum glucose, triglyceride, and total cholesterol in BM-transplanted *Ldlr*^{-/-} mice at different times after HFD feeding; n=6-16 mice per genotype and time-point. (d) Intima area in the aortic sinus of BM-transplanted *Ldlr*^{-/-} mice fed a HFD for 8 or 12 weeks; n=6-12 mice per genotype and time-point. (e) Representative images of H&E-stained transverse sections of aortic sinus from BM-transplanted *Ldlr*^{-/-} mice fed a HFD and scored on the Stary classification from I to VI (Stary et al., 1995); scale bar, 200 μ m. Data were tested by two-way ANOVA followed by Bonferroni's post-test.

Reviewers' comments:

Reviewer #1 (Remarks to the Author):

Thank you for not only clarifying the issues I raised, but in presenting a significant amount of new data.

Reviewer #2 (Remarks to the Author):

The in vivo studies utilizing the CCR5 antagonist are interesting and the phenotype is striking. However, several concerns remain. First, while an attempt at a clinical Stary classification has been made, lesion size should also be reported. Second, there is no evidence that patrolling macrophages have been recruited to the 'at risk' area of increased lipid uptake. After only 3d of HFD, the 'at risk' area is defined by a few resident intimal myeloid cells in the lesser curvature of the aortic arch (Cybulsky and colleagues JEM, 2009). It is unclear to this reviewer whether the CD115+ cells described in Figure 5b associate with this region. Also, a difference of 3 cells/field versus 2 is underwhelming. While possibly statistically significant, the authors should verify the biological significance of this finding to atheroma development.

The 8 week HFD data in Figure 8 is also interesting, but does not directly link the MT4-deficiency phenotype with Ly6Clow patrolling monocyte recruitment. MRV delivery abolishes monocyte recruitment/proliferation in lesions of control (vehicle) mice as well. While this data set (and previously published studies) links CCR5 to Ly6Clow monocyte recruitment in atherogenesis, it doesn't increased monocyte recruitment in MT4-deficient animals may occur independently of CCR5.

My reference to adoptive transfer experiments was not clearly defined. I was in fact, referring to the bone marrow transplant studies outlined in Figures 2 and now Figures 5 and 6. It is imperative that the authors provide steady state immune cell profiles 6-8 weeks after transplant, but prior to high fat feeding. The hematopoietic stem and progenitor cell (and possibly other white blood cell) profiles may differ between experimental groups due to differences in seeding potential of HSPC in MT4 KO mice. For example, are there equivalent percentages and numbers of circulating Ly6Clow monocytes between experimental groups 6-8 weeks after transplantation?

While this reviewer is convinced that MT4-MMP mediates Ly6Clow patrolling behaviour, insufficient proof has been provided to substantiate the claim that MT4-MMP-mediated effects on atherosclerosis are DIRECTLY related to Ly6Clow monocyte recruitment.

POINT-BY-POINT REPLY LETTER

Reviewers' comments:

Reviewer #1 (Remarks to the Author):

Thank you for not only clarifying the issues I raised, but in presenting a significant amount of new data.

We are grateful to the reviewer's positive comment and appreciation of the effort and the novel information included in this revised version.

Reviewer #2 (Remarks to the Author):

A. The in vivo studies utilizing the CCR5 antagonist are interesting and the phenotype is striking. However, several concerns remain. First, while an attempt at a clinical Stary classification has been made, lesion size should also be reported. Second, there is no evidence that patrolling macrophages have been recruited to the 'at risk' area of increased lipid uptake. After only 3d of HFD, the 'at risk' area is defined by a few resident intimal myeloid cells in the lesser curvature of the aortic arch (Cybulsky and colleagues JEM, 2009). It is unclear to this reviewer whether the CD115+ cells described in Figure 5b associate with this region. Also, a difference of 3 cells/field versus 2 is underwhelming. While possibly statistically significant, the authors should verify the biological significance of this finding to atheroma development.

We appreciate the positive comment from the reviewer on the interest of the in vivo experiment and the data obtained with the CCR5 antagonist (Maraviroc, MRV). We have addressed his/her remaining concerns as follows:

1. The lesion size was already included in the previous version assessed by Red Oil-positive area (see previous Supp Figure 11b) but as requested we have now complemented this analysis with the quantification of the intima area in H&E-stained sections from aortas in the MRV experiment. As shown in the revised Supp Figure 11, MRV treatment significantly decreased the intima area in both wild-type and MT4-MMP-null BMT Ldlr^{-/-} mice fed a HFD for 8 weeks with no further differences observed between genotypes. These data are in line with the Red Oil quantification (now included as revised Figure 5e).
2. We have now specified that indeed the analysis of early patrolling monocyte (PMo) recruitment in Figures 5a and b was performed in the athero-prone lesser curvature of the aorta. And we have included image insets of CD31 staining which show polygonal and non-aligned endothelial cells typical of areas subjected to disturbed flow as the lesser curvature (revised Figure 5b).
3. We agree the numbers of patrolling monocytes (PMos) recruited are particularly low in the vehicle-treated BMT Ldlr^{-/-} mice (Figure 8a) but still within the range of what it has previously been reported (Tacke et al, JCI, 2007). The biological significance of these findings comes from the fact that eliminating this increment of just 1-2 PMo at the early stage thus reaching equivalent numbers of early recruited PMo in mice transplanted with WT and MT4-MMP-null bone marrow cells results in the abolition of the accelerated AT phenotype in the latter. This argues in favor of increased PMo recruitment as a main contributor of the subsequent AT phenotype in the absence of MT4-MMP. If other mechanisms following recruitment were the responsible, the reduction in early recruitment may impact macrophage number/composition at early stages (7 days) but still result in increased macrophage burden and lipid deposits at later stages (8 weeks) in the MT4-MMP-null BMT Ldlr^{-/-} mice. We have now mentioned this point in the revised discussion section (page 15-16 lines 355-358).

B. The 8 week HFD data in Figure 8 is also interesting, but does not directly link the MT4-deficiency phenotype with Ly6Clow patrolling monocyte recruitment. MRV delivery abolishes monocyte recruitment/proliferation in lesions of control (vehicle) mice as well. While this data set (and previously published studies) links CCR5 to Ly6Clow monocyte recruitment in atherogenesis, it doesn't increased monocyte recruitment in MT4-deficient animals may occur independently of CCR5.

The reviewer critically points that MRV abolishes monocyte recruitment in both control (the reviewer here says vehicle but he/she should referred in wild-type mice treated with MRV) and MT4-MMP-null BMT Ldlr^{-/-} mice. But indeed this MRV effect has previously been reported (Cipriani et al., Circulation 2013) and again argues in favor of the relevance (still poorly characterized) of early recruitment of PMo to the overall AT phenotype. Regarding the last sentence although unclear, the reviewer seems to claim that this approach does not prove whether the increased monocyte recruitment may be

related to other CCR5 independent pathways. The strength and beauty of the MRV experiment in this context is that it eliminates the enhancement of PMo recruitment in the absence of MT4-MMP making the early starting recruitment equivalent in WT and MT4-MMP-null BMT *Ldlr*^{-/-} mice and demonstrating that in this case there is no subsequent acceleration of atherosclerosis after 8 weeks of HFD. These data proved that the enhanced early PMo recruitment contributes and is required for the AT phenotype in the absence of MT4-MMP. Of course we agree, and the MRV data in Figure 8a show, that there are a few monocytes in the aorta at 3 days of high fat diet even in MRV-treated mice that could be related to the presence of aortic resident immune cells, the non-100% efficiency of MRV inhibition or the contribution of other adhesion receptors for example the $\alpha 4$ integrin as recently reported in the atherogenic arterial context (Quintar et al., *Circ Res* 2017). In any case the possible CCR5 independent pathways do not seem to contribute to the observed AT phenotype.

C. My reference to adoptive transfer experiments was not clearly defined. I was in fact, referring to the bone marrow transplant studies outlined in Figures 2 and now Figures 5 and 6. It is imperative that the authors provide steady state immune cell profiles 6-8 weeks after transplant, but prior to high fat feeding. The hematopoietic stem and progenitor cell (and possibly other white blood cell) profiles may differ between experimental groups due to differences in seeding potential of HSPC in MT4 KO mice. For example, are there equivalent percentages and numbers of circulating Ly6Clow monocytes between experimental groups 6-8 weeks after transplantation?

While we appreciate that this point may be of interest, we regret that the reviewer did not ask for these data in his/her original review. We had already mentioned in the text that there were no differences in circulating monocytes and shown in previous Figure 3a that only the % of circulating PMo in steady-state conditions was reduced 4 weeks after BMT. In order to check for possible differences in seeding potential of HSPC in MT4 KO mice as requested by the reviewer, we have now included the % of different populations in the bone marrow and blood of mice 4 weeks after bone marrow transplant (instead of after 6-8 weeks) a time-point at which reconstitution is complete and donor/host chimerism in bone marrow-derived cells is lower making the results about seeding more robust (Miller, *J Hemother and Stem Cell Res*, 2002). We have thus quantitated based on flow cytometry analysis (Practical Flow Cytometry in Haematology Diagnosis, 2013) the % of progenitors and of mature progeny in the myeloid and lymphoid lineages in the bone marrow and blood. As included in revised Supp. Figure 2b there are no significant differences in progenitors and mature leukocytes either in the bone marrow or blood between wild-type and MT4-MMP-null bone marrow-transplanted *Ldlr*^{-/-} mice. These data argue in favor of a specific impact of MT4-MMP deficiency on circulating PMOs and against defects in HSPC seeding. Moreover, the fact that double *Ldlr*^{-/-}/MT4-MMP knock-out mice fed a high fat diet display a similar AT acceleration phenotype (Supp Figure 3) further support that the AT phenotype is not related to defects in MT4-MMP-null bone marrow engraftment after transplant (although this procedure may exacerbate AT development as previously noticed; Randolph, *Circ Res* 2014).

While this reviewer is convinced that MT4-MMP mediates Ly6Clow patrolling behaviour, insufficient proof has been provided to substantiate the claim that MT4-MMP-mediated effects on atherosclerosis are DIRECTLY related to Ly6Clow monocyte recruitment.

The MRV experiment demonstrates that the enhanced PMo recruitment in the absence of MT4-MMP is required and contributes to the AT phenotype but as we clearly state throughout the manuscript (from the title, to the Abstract and the Discussion), this is the key promoting mechanism but PMo-derived MT4-MMP-null macrophages also behave differently (resisting apoptosis better and up-taking more lipids) what also contributes to the AT phenotype. This was mentioned in the previous and this re-revised version.

REVIEWERS' COMMENTS:

Reviewer #1 (Remarks to the Author):

The authors have strengthened their manuscript further and I agree with them that the concerns of the other reviewer are strongly addressed. I think there are two points they discuss in their responses that need to be highlighted in the final version, both related to the Tacke 2007 paper they cite.

1) In that paper, it was shown that CCR5 was important for the recruitment of the LY6Clow cells, but was not the only factor. While the authors make a good case that it is the CCR5 component that is relevant to their studies, it is an important biological finding that should be highlighted- and that it remains to determine (by others) what the non-CCR5 factors are and what their functional significance is.

2) The other point is that monocyte recruitment studies by almost any method is notoriously inefficient, so the low number of cells reported are not surprisingly. If this were the only line of evidence to support their model, I would agree with the other reviewer that this is not a robust finding. In the context of the entire slate of studies, however, it is not a "deal breaker". Perhaps the best way to express the results of this experiment is to acknowledge up front in the published version that low efficiency is a common limitation in these sorts of assays (and cite a few examples, such as Tacke et al 2007), and that the results are consistent with decreased recruitment.

POINT-BY-POINT REPLY LETTER

Reviewers' comments:

Reviewer #1 (Remarks to the Author):

The authors have strengthened their manuscript further and I agree with them that the concerns of the other reviewer are strongly addressed. I think there are two points they discuss in their responses that need to be highlighted in the final version, both related to the Tacke 2007 paper they cite.

1) In that paper, it was shown that CCR5 was important for the recruitment of the LY6Clow cells, but was not the only factor. While the authors make a good case that it is the CCR5 component that is relevant to their studies, it is an important biological finding that should be highlighted and that it remains to determine (by others) what the non-CCR5 factors are and what their functional significance is.

We appreciate this comment and agree with the reviewer in the interest of addressing the contribution of non-CCR5 factors to the recruitment of patrolling monocytes to the atherosclerotic plaque. Accordingly, we have added a sentence highlighting this issue in the revised Discussion section (lines 338-340).

2) The other point is that monocyte recruitment studies by almost any method is notoriously inefficient, so the low number of cells reported are not surprisingly. If this were the only line of evidence to support their model, I would agree with the other reviewer that this is not a robust finding. In the context of the entire slate of studies, however, it is not a "deal breaker". Perhaps the best way to express the results of this experiment is to acknowledge up front in the published version that low efficiency is a common limitation in these sorts of assays (and cite a few examples, such as Tacke et al 2007), and that the results are consistent with decreased recruitment.

We concur with the reviewer and appreciate his/her observation about the limited numbers of patrolling monocytes present at early lesions. As the reviewer mentioned, in our studies we have used several and complementary approaches to analyze this point and we have always obtained low patrolling monocyte numbers in early lesions within the range previously published by other groups. Following the reviewer's suggestion we have rephrased the Results section and included two sentences (lines 215-216 and 278-280) acknowledging up front this limitation of the assay.